# The genetic basis of aneuploidy tolerance in wild yeast

**James Hose[1], Leah E Escalante[1,2], Katie J Clowers[2†], H Auguste Dutcher[1,2], DeElegant Robinson[1], Venera Bouriakov[1,3], Joshua J Coon[1,3,4,5,6], Evgenia Shishkova[1,6], Audrey P Gasch[1,2,3]\***

[1]Center for Genomic Science Innovation, University of Wisconsin–Madison, Madison, United States; [2]Laboratory of Genetics, University of Wisconsin-Madison, Madison, United States; [3]Great Lakes Bioenergy Research Center, Madison, United States; [4]Department of Biomolecular Chemistry, University of Wisconsin–Madison, Madison, United States; [5]Department of Chemistry, University of Wisconsin–Madison, Madison, United States; [6]Morgridge Institute for Research, Madison, United States

**Abstract** Aneuploidy is highly detrimental during development yet common in cancers and pathogenic fungi – what gives rise to differences in aneuploidy tolerance remains unclear. We previously showed that wild isolates of *Saccharomyces cerevisiae* tolerate chromosome amplification while laboratory strains used as a model for aneuploid syndromes do not. Here, we mapped the genetic basis to Ssd1, an RNA-binding translational regulator that is functional in wild aneuploids but defective in laboratory strain W303. Loss of *SSD1* recapitulates myriad aneuploidy signatures previously taken as eukaryotic responses. We show that aneuploidy tolerance is enabled via a role for Ssd1 in mitochondrial physiology, including binding and regulating nuclear-encoded mitochondrial mRNAs, coupled with a role in mitigating proteostasis stress. Recapitulating *ssd1Δ* defects with combinatorial drug treatment selectively blocked proliferation of wild-type aneuploids compared to euploids. Our work adds to elegant studies in the sensitized laboratory strain to present a mechanistic understanding of eukaryotic aneuploidy tolerance.

\*For correspondence:
agasch@wisc.edu

Present address: †Ginkgo Bioworks, Boston, United States

Competing interests: The authors declare that no competing interests exist.

## Introduction

Aneuploidy, in which cells carry an abnormal number of one or more chromosomes, is highly detrimental during mammalian development, since amplification of most human chromosomes is inviable during embryogenesis. Imbalanced and especially elevated expression from altered chromosomes is thought to tax cellular proteostasis, both by producing too much protein and when stoichiometric imbalance of interacting proteins disrupts cooperative folding (*Donnelly and Storchová, 2015*; *Oromendia and Amon, 2014*; *Pavelka and Rancati, 2013*). Yet ≥90% of tumors are aneuploid with little detriment and even benefits to cells, and the degree of aneuploidy is associated with poorer patient prognosis (*Holland and Cleveland, 2012*; *Targa and Rancati, 2018*). Aneuploidy is also common in several fungal species including fungal pathogens. In fact, chromosome amplification represents a major route to drug resistance in pathogenic infections, when amplification of drug transporters and defense mechanisms promotes drug evasion (*Wertheimer et al., 2016*; *Ni et al., 2013*; *Bennett et al., 2014*). Why aneuploidy is benign or beneficial in some cells but highly deleterious in others is not understood.

The yeast *Saccharomyces cerevisiae* has been a formidable model to understand why chromosome amplification is toxic in the first place. Several studies characterized suites of aneuploid laboratory strains to understand the mechanisms of aneuploidy toxicity and the effects of chromosomal amplification. In a well-studied laboratory strain, chromosome amplification leads to reduced cell

growth, metabolic alterations, altered cell-cycle progression in part through aberrant cyclin regulation, activation of a common transcriptome program regardless of the amplified chromosome, and signatures of protein aggregation and defects clearing misfolded peptides, referred to as proteostasis stress (*Torres et al., 2007*; *Torres et al., 2010*; *Oromendia et al., 2012*; *Sheltzer et al., 2012*; *Thorburn et al., 2013*; *Dephoure et al., 2014*; *Dodgson et al., 2016*; *Brennan et al., 2019*). Despite the deleterious effects reported in lab strains, chromosome amplification is beneficial in the right environment and provides a rapid route to phenotypic evolution (*Rancati et al., 2008*; *Pavelka et al., 2010*; *Yona et al., 2012*; *Filteau et al., 2015*; *Fontanillas et al., 2010*). This is consistent with the prevalence of chromosome amplification in fungal pathogens emerging after drug-treatment regimens (*Ni et al., 2013*; *Selmecki et al., 2009*; *Selmecki, 2006*).

Studies in laboratory strains have clearly generated important information on the causes and consequences of aneuploidy. However, we previously reported a striking difference among wild isolates: a substantial number of wild strains are naturally aneuploid, in some cases carrying extra copies of multiple chromosomes (*Gasch et al., 2016*; *Hose et al., 2015*). Recent large-scale sequencing efforts provide confirmatory evidence, reporting over 20% of sequenced strains as aneuploid, with each of the 16 yeast chromosomes represented across affected strains (*Peter et al., 2018*). In contrast to well-studied laboratory strain W303, naturally aneuploid yeast show only subtle growth defects, no detectable metabolic differences, and lack evidence of the canonical stress response (*Gasch et al., 2016*; *Hose et al., 2015*). The relative tolerance is not a result of adaptation: we showed that naturally euploid strains selected for chromosome amplification also showed relatively mild growth defects, and euploid derivatives of aneuploid isolates grew similarly to the aneuploid parent. Although some strains show variable karyotypes over time, picking up or losing chromosomes during division, chromosome amplification in other strains is generally stable (*Gasch et al., 2016*). Thus, many wild yeast strains tolerate chromosome amplification whereas W303 cannot.

Here, we mapped the genetic basis for this phenotypic difference, by crossing a naturally aneuploid strain isolated from oak soil, YPS1009 with extra copies of Chromosome XII (Chr12), to laboratory strain W303 carrying an extra copy of Chr12. Mapping and confirmatory genetics reveal that the basis for the difference in aneuploidy tolerance lies in *SSD1*, encoding an RNA binding protein known to be hypomorphic in W303. Our results point to combinatorial dysfunction in mitochondrial physiology and cytosolic protein homeostasis as the basis for aneuploidy toxicity in *ssd1-* strains, and in wild-type aneuploids with drug-induced defects. Integrating our results with past yeast and mammalian studies presents a holistic view of eukaryotic responses to chromosome amplification.

## Results

To identify the genetic basis of differential aneuploidy tolerance, we crossed a haploid derivative of oak-soil strain YPS1009 disomic for chromosome 12 (YPS1009_Chr12) to W303 disomic for the same chromosome (W303_Chr12, *Figure 1A*). Haploid F2 segregants all harbor two copies of Chr12 but display quantitatively different growth rates (*Figure 1—figure supplement 1*). To score aneuploidy sensitivity, we focused on W303_Chr12 phenotypes, namely small colony size, slow growth, and/or propensity of the culture to lose the amplified chromosome during passaging. We realized during tetrad dissection that W303-inherited auxotrophies, especially adenine auxotrophy, influenced aneuploidy tolerance (*Figure 1—figure supplement 2A-B*). We therefore selected an F2 segregant prototrophic for influential markers (called 'sp100'), backcrossed it to the tolerant YPS1009_Chr12 parent, and scored aneuploidy sensitivity as above (*Figure 1A*) to generate pools of aneuploidy-sensitive and aneuploidy-tolerant segregants (*Figure 1—figure supplement 2C-D*). To control for other genetic influences on growth rate and/or colony size, we also performed a control cross of the euploid parents, generating pools of euploid segregants with small versus large colony sizes (see Materials and methods).

Bulk analysis of aneuploidy-sensitive and -tolerant backcrossed segregants revealed a major-effect locus on Chr4 that was nearly fixed for W303 alleles in the aneuploidy-sensitive pool (*Figure 1B* and *Figure 1—figure supplements 3*, *4*) but not small colonies from a euploid-control cross (*Figure 1C*, see Materials and methods). The locus spanned *SSD1*, encoding an RNA-binding protein. This locus harbors a premature stop codon in W303 that deletes 44% of the Ssd1 protein including conserved RNA binding domains (*Uesono et al., 1997*; *Sutton et al., 1991*). Ssd1 is best characterized for regulating localization and translation of cell-wall destined mRNAs, delivered by

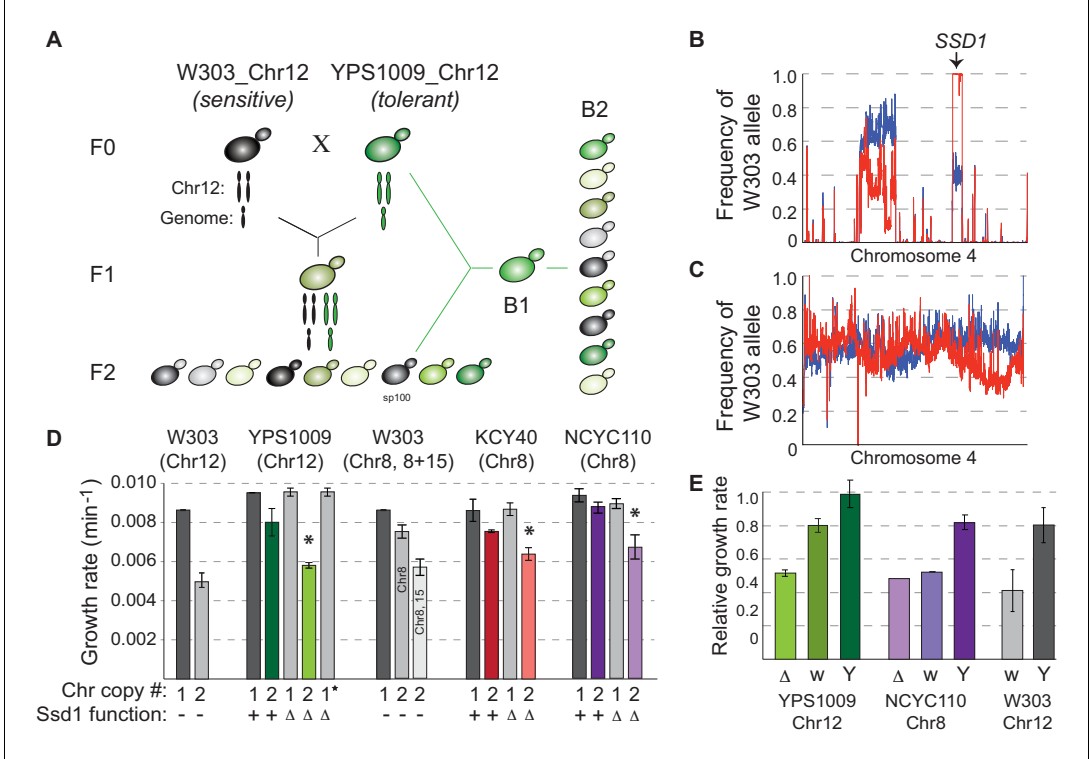

**Figure 1.** *SSD1* is required for aneuploidy tolerance. (**A**) Mapping schema, see Materials and methods. (**B**) W303 allele frequency across Chr4 in the pool of aneuploidy-sensitive (red) versus -tolerant (blue) B2 segregants or C) small (red) versus large (blue) colony pools from the euploid-control cross. (**D**) Average and standard deviation of growth rates for denoted strains with amplified chromosomes, indicated above. Number of chromosomes per haploid genome, *SSD1* status (Δ, deletion; –, *ssd1*$^{W303}$), and star indicating euploid revertant are indicated below. Asterisk, p<0.005, T-test comparing aneuploids with and without *SSD1*. (**E**) Average and standard deviation of growth of aneuploid *ssd1-* strains harboring empty vector (Δ), *ssd1*$^{W303}$ (w) or *SSD1*$^{YPS1009}$ (Y), relative to the isogenic aneuploid wild type with empty vector (or euploid cells with empty vector in the case of W303 *ssd1*$^{W303}$ cells). The online version of this article includes the following source data and figure supplement(s) for figure 1:

**Source data 1.** Source data for *Figure 1*.
**Figure supplement 1.** Aneuploidy tolerance varies in W303 and YPS1009 strains.
**Figure supplement 2.** Auxotrophies influence aneuploidy tolerance.
**Figure supplement 3.** Multipool output for initial cross.
**Figure supplement 4.** Multipool output for the backcross.
**Figure supplement 5.** Ssd1 is required for aneuploidy tolerance in diploid YPS1009.

Ssd1 to the growing bud during active growth but to P-bodies for translational silencing following stress or mitotic defects (*Jansen et al., 2009*; *Kurischko et al., 2011a*). Ssd1 has also been implicated in a large number of suppressor screens and has a role in aging and quiescence (*Miles et al., 2019*; *Li et al., 2013*; *Hu et al., 2018*). W303 carries a premature stop codon that ablates RNA binding domains, which underlies several phenotypic differences reported between W303 and other strains (*Kaeberlein et al., 2004*; *Moriya and Isono, 1999*; *Ohyama et al., 2010*; *Uesono et al., 1994*).

Genetic analysis confirmed that *SSD1* underlies the difference in aneuploidy tolerance. *SSD1* deletion had little effect on the growth of euploid YPS1009 but significantly retarded YPS1009_Chr12 proliferation, comparable to W303_Chr12 (*Figure 1D*). The phenotype was true in both haploid and diploid versions of the strain (*Figure 1—figure supplement 5*). Growth rate was restored if cells lost the extra chromosome during passaging (*Figure 1D*, star) or if *SSD1*$^{YPS1009}$ was reintroduced (*Figure 1E*). To test if Ssd1's role was exclusive to this genetic background or chromosome amplification, we deleted *SSD1* in naturally aneuploid, diploid West African strain tetrasomic for Chr 8 (NCYC110_Chr8) and in a derived aneuploid vineyard strain, KCY40_Chr8 (*Hose et al., 2015*). *SSD1* deletion sensitized cells to chromosome amplification, showing that the effect is independent of

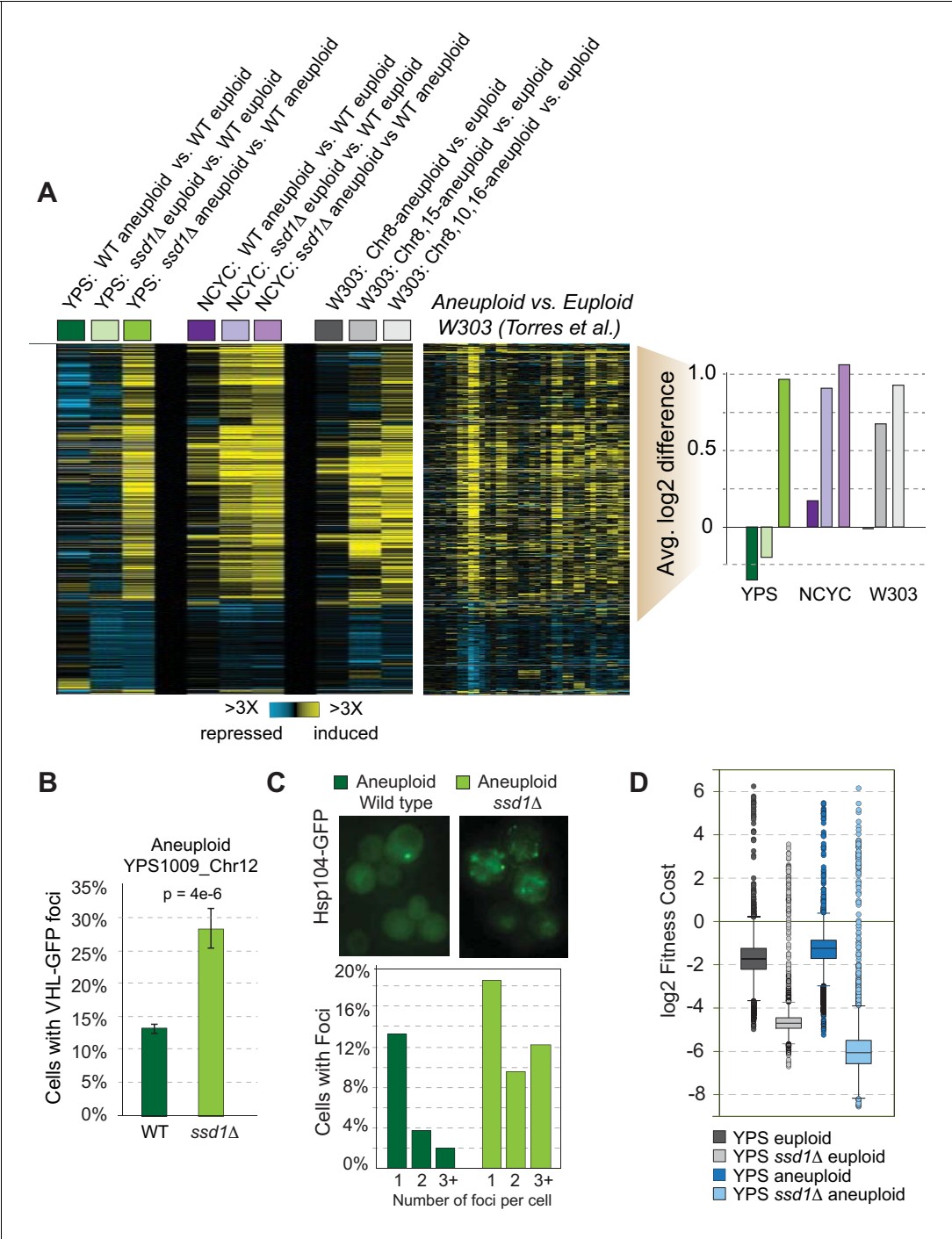

**Figure 2.** *SSD1* deletion induces aneuploidy signatures. (A) Replicate-averaged log$_2$ expression differences for strain comparisons (columns) across 861 genes (rows) differentially expressed in mutant versus wild-type aneuploids, see text. Strains include haploid YPS1009 disomic for Chr12, diploid NCYC110 tetrasomic for Chr 8, or haploid W303 derivatives with different chromosome amplifications. Corresponding data from Torres et al. and average log$_2$ differences in expression of induced genes are shown, where colors indicate strain labels from left. (B–C) Quantification of B) VHL-GFP foci and C) Hsp104-GFP in aneuploid strains. Data represent average and standard error of the mean (SEM) across biological triplicates, p from Fisher's exact test. (D) Distribution of replicate-averaged fitness costs from high-copy plasmid over-expression in each strain (see Materials and methods). The online version of this article includes the following source data and figure supplement(s) for figure 2:

**Source data 1.** Transcriptome data shown in *Figure 2A*.
**Figure supplement 1.** Hsp104-GFP foci in euploid cells.

genetic background and duplicated chromosome (and is thus also independent of the rDNA locus on Chr 12) (*Figure 1D*). Reintroducing the YPS1009 allele of *SSD1* complemented the aneuploidy sensitivity of multiple strain backgrounds (*Figure 1E*), whereas the W303 allele provided no complementation in the NCYC110_Chr8 *ssd1Δ* strain and partial complementation in YPS1009_Chr12 *ssd1Δ* (*Figure 1D*). Importantly, expressing the YPS1009 allele in W303_Chr12 largely corrected its sensitivity (with a remaining contribution likely from the adenine auxotrophy, see *Figure 1—figure supplement 2B*), demonstrating that the *ssd1$^{W303}$* allele is responsible for aneuploidy sensitivity in W303. Thus, Ssd1 plays a generalizable role in tolerating chromosome amplification across multiple strains and chromosome duplications.

## Loss of Ssd1 recapitulates multiple signatures of aneuploid W303

W303 studies reported a transcriptomic signature of aneuploidy independent of amplified chromosome identity, but this is not seen in wild aneuploid strains (*Torres et al., 2007*; *Hose et al., 2015*). To test dependence on Ssd1, we followed transcriptomes of natural aneuploids and their *ssd1Δ* derivatives, with or without extra chromosomes. We identified 861 genes with altered expression in both YPS1009_Chr12 *ssd1Δ* and NCYC110_Chr8 *ssd1Δ* mutants compared to their isogenic wild-type aneuploids (false discovery rate, FDR < 0.05, *Figure 2A*). Induced genes showed little change in euploid YPS1009 *ssd1Δ* but were up-regulated when *SSD1* was deleted in the context of Chr12 amplification. NCYC110 showed similar trends, except that in this strain we observed a weak expression signature in euploid *ssd1Δ* cells that was exacerbated when Chr8 was amplified. Repressed genes included rRNA and tRNA processing and cytokinesis factors, whereas induced genes encompassed the environmental stress response (ESR *Gasch et al., 2000*), oxidoreductases, carbohydrate and energy metabolism, and genes involved in mitochondrial degradation (p<1e-4, hypergeometric test). This response effectively recapitulates the expression signature seen in W303 aneuploids (*Torres et al., 2007*, *Figure 2A*). The response was exacerbated with increasing DNA content in W303 carrying multiple extra chromosomes (*Figure 2A*). Thus, the previously reported aneuploidy transcriptome signature results from defective Ssd1 function, independent of affected chromosome, exacerbated with additional DNA content, and with some strain-specific nuances.

We wondered if other aneuploidy signatures seen in W303 could be explained by defective Ssd1. In addition to growth delay, aneuploid W303 strains reportedly exhibit larger cell size, altered cyclin Cln2 abundance, delayed G1/S progression, metabolic defects, and signatures of proteotoxicity including protein aggregation and a defect degrading misfolded protein (*Torres et al., 2007*; *Oromendia et al., 2012*; *Thorburn et al., 2013*). While these seminal studies generated important information on aneuploidy toxicity in a sensitized strain, many phenotypes likely result from defective Ssd1. YPS1009_Chr12 *ssd1Δ* grows slower (*Figure 1*), produces ~33% higher optical versus cell density indicating larger size, and displays metabolic defects encompassing defective respiratory growth (see Figure 4). We confirmed that Ssd1 binds many cell-cycle transcripts including *CLN2* (*Supplementary file 2*), which is translationally regulated by Ssd1 and explains altered G1/S progression (*Ohyama et al., 2010*).

Loss of *SSD1* also explains proteotoxicity observed for aneuploid strains. First, we followed accumulation of human Von Hippel Lindau (VHL) protein upon over-expression. Because VHL cannot fold in the absence of its interacting proteins in yeast, accumulation of VHL-GFP foci reflects misfolded protein that has yet to be cleared by the proteasome (*McClellan et al., 2005*; *Kaganovich et al., 2008*). We found that significantly more YPS1009_Chr12 *ssd1Δ* cells accumulated VHL-GFP foci compared to wild type (*Figure 2B*). We also followed the protein disaggregase Hsp104, which binds misfolded and aggregated proteins in discrete protein quality-control centers (*Kaganovich et al., 2008*; *Chernova et al., 2017*). Wild-type aneuploids showed no obvious difference in the number of Hsp104-GFP foci compared to euploids (*Figure 2C* and *Figure 2—figure supplement 1*), indicating that gross protein aggregation is not a universal feature of aneuploid yeast. However, mutant aneuploids lacking *SSD1* showed a higher proportion of cells with Hsp104-GFP foci, and more foci within those cells, compared to wild-type aneuploids (*Figure 2C*). Ssd1 was previously implicated in protein homeostasis after heat shock (*Mir et al., 2009*), but our results demonstrate that chromosome amplification in the absence of other stresses is enough to provoke misfolding in *ssd1Δ* cells. Together, these results show that myriad signatures of W303 aneuploidy can occur due to defective Ssd1.

Ssd1 mutants could be sensitive to specific genes on the amplified chromosomes, or they could have a generalized sensitivity to the burden of extra DNA/protein. To distinguish these models, we transformed YPS1009 strains with a barcoded, high-copy gene over-expression library and measured relative fitness costs after 5 generations of growth (see Materials and methods). Both the euploid and aneuploid *ssd1Δ* mutants were highly sensitive to the library (*Figure 2D*): genes that were deleterious in wild type were toxic in the mutant, while many genes with neutral effect in parental strains were deleterious in the absence of *SSD1*. We cannot exclude a defect maintaining the high-copy 2-micron plasmid (indeed, the mutant cells do not grow well with the empty vector, and Ssd1 is already implicated in plasmid maintenance *Uesono et al., 1994*). Nonetheless, both the euploid and aneuploid *ssd1Δ* mutants are highly sensitive to the 2-micron overexpression library.

## Ssd1 binds RNAs and alters aneuploid proteomes

We focused on YPS1009 strains to elucidate Ssd1 function in aneuploidy tolerance. Revisiting the YPS1009_Chr12 *ssd1Δ* transcriptome revealed broader induction of genes, including mRNAs whose proteins localize to subcellular compartments such as mitochondria, ER, vacuole, peroxisome, plasma membrane, and the cell wall (p<1e-4, hypergeometric test). Many of these organelles functionally and physical interact (*Scorrano et al., 2019*), raising the possibility of broader inter-organelle

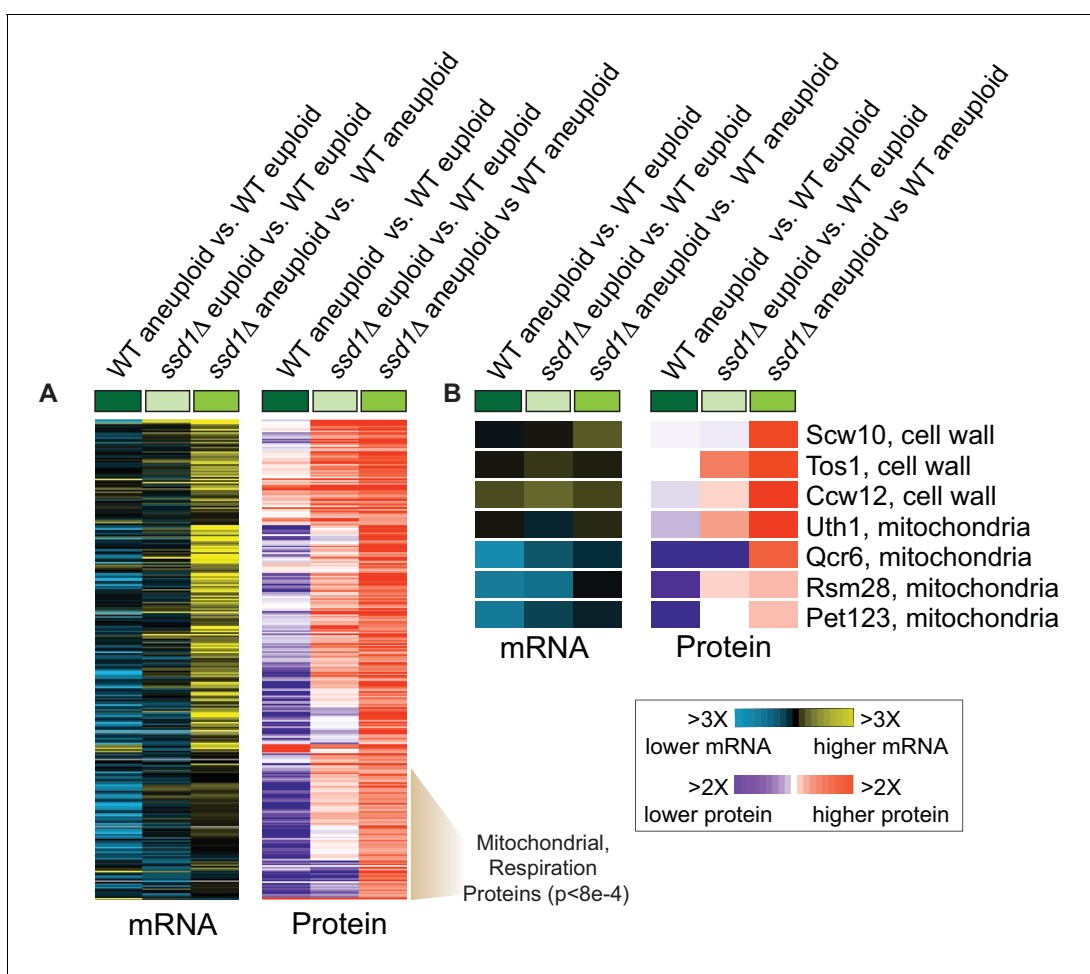

**Figure 3.** Ssd1 affects the proteome in aneuploid YPS1009 cells. (A) Replicate-averaged log2(fold difference) in abundance across 301 significant proteins (FDR < 0.05) and their corresponding mRNAs (rows) for denoted comparisons (columns), where colors represent the magnitude of change according to the key. The indicated cluster is enriched for mitochondrial proteins and respiration factors (hypergeometric test). (B) Representative Ssd1-bound transcripts from (A).

The online version of this article includes the following source data for figure 3:

**Source data 1.** source data for *Figure 3*.

issues. Consistent with this notion, the *ssd1Δ* aneuploid also showed transcriptional signatures of the unfolded ER-protein response (*Travers et al., 2000*) and mitochondrial protein import stress (*Weidberg and Amon, 2018*) (see Materials and methods).

To test if Ssd1 binds a broader set of mRNAs, we sequenced RNAs recovered from Ssd1 immuno-precipitation (see Materials and methods). The 286 associated mRNAs (FDR < 0.05, *Supplementary file 2*) were heavily enriched for transcripts encoding cell-wall proteins as expected, but the group was also enriched for cell-cycle regulated mRNAs (including cyclins *CLN2* and *CLB2/4/5*) and those involved in budding, RNA metabolism, and sterol transport (p<1e-4). Myriad other functions were also represented, such as proteins in chromatin regulation, transcription, lipid biogenesis, endocytosis, protein homeostasis, and mitochondrial function, some previously noted (*Jansen et al., 2009*; *Hogan et al., 2008*). Interestingly, Ssd1 also bound mRNAs encoding osmotic-response regulators (*SLN1, SHO1, MSB2, HOT1*), notable since osmotic stress was recently implicated in aneuploidy responses of a different lab strain (*Tsai et al., 2019*). Ssd1-bound mRNAs were not enriched for those encoded on the amplified chromosome, nor transcripts disproportionately expressed compared to DNA content (*Hose et al., 2015*), suggesting that the mechanism of aneuploidy tolerance is not through modulation of amplified-gene expression. Most (60%) bound transcripts were not differentially expressed in the aneuploid mutant, downplaying a generalizable role in regulating mRNA abundance. In turn, most mRNAs differentially expressed in the aneuploid mutant are not Ssd1-bound, suggesting widespread secondary responses to the primary defect(s).

Since Ssd1 regulates translation via direct RNA binding (*Jansen et al., 2009*; *Wanless et al., 2014*; *Kurischko et al., 2011b*), we next used quantitative proteomics to measure effects on the cellular proteome. 301 of 3906 measured proteins were more abundant in the aneuploid mutant versus wild type (FDR < 0.05, *Figure 3A*). Many emerge from induced transcripts; however, a large fraction of proteins was elevated beyond mRNA differences, including cell-wall proteins, nuclear-encoded mitochondrial proteins (p=6e-4, hypergeometric test), and others. Many of these proteins were also elevated in the euploid mutant without significant mRNA changes (*Figure 3A*). Although there was no enrichment for proteins encoded by Ssd1-bound transcripts, several elevated proteins emerge from Ssd1 targets, including cell-wall transcripts and several nuclear transcripts encoding mitochondrial proteins. For example, *UTH1* encoding a mitochondrial protein linked to aging is bound by Ssd1 in our and other studies and is known to be translationally regulated by direct Ssd1 binding at specific mRNA locations (*Wanless et al., 2014*; *Camougrand et al., 2004*; *Camougrand et al., 2000*). Although protein induction was evident in euploid *ssd1Δ* cells, the defect was exacerbated by Chr12 amplification, with ~4X more Uth1 protein despite little difference in mRNA (*Figure 3B*). Other mitochondrial proteins emanating from Ssd1-bound mRNAs were also significantly elevated in the *ssd1Δ* aneuploid, including several mitochondrial ribosomal proteins (*Figure 3B*). Thus, Ssd1 affects the proteome of aneuploid cells, including from bound transcripts.

## Ssd1 is important for mitochondrial function and inheritance

Our past work revealed that wild aneuploid strains down-regulate many nuclear encoded mitochondrial transcripts, a response also seen in Down syndrome models (*Helguera et al., 2013*; *Liu et al., 2017*), hinting that mitochondrial regulation is important for tolerating chromosome amplification (*Hose et al., 2015*). In the current work, multiple lines implicated mitochondrial effects in *ssd1Δ* aneuploids. To explore this, we tested mitochondrial function in YPS1009 strains. We found a synergistic defect between *SSD1* deletion and aneuploidy when cells experienced mitochondrial stress. YPS1009_Chr12 *ssd1Δ*, NCYC110_Chr8 *ssd1Δ* (*Figure 4A*), and aneuploid W303 (*Hose et al., 2015*) were all sensitive to non-fermentable acetate, beyond what is expected from the additive effects of aneuploidy and reduced respiratory growth rate. This demonstrates Ssd1-dependent respiratory dysfunction autonomous of strain background or affected chromosome. YPS1009_Chr12 *ssd1Δ* cells were also susceptible to sub-lethal doses of carbonyl-cyanide 3-chlorophenylhydrazone (CCCP), which uncouples mitochondrial membrane potential (*Figure 4B*). Notably, aneuploid *ssd1Δ* cells were no more sensitive than expected to cell wall or ER stress (*Figure 4—figure supplement 1*), indicating a specific interaction with mitochondrial challenge. In the process of this work, we discovered that *ssd1Δ* aneuploid cells also showed a striking difference in mitochondrial morphology. Wild-type YPS1009_Chr12 grew well but harbored many globular mitochondria compared to the euploid tubular shape (*Figure 4C*). Although the impact of this morphology is not clear, YPS1009_Chr12 *ssd1Δ* displayed significant differences, including more tubular forms and increased

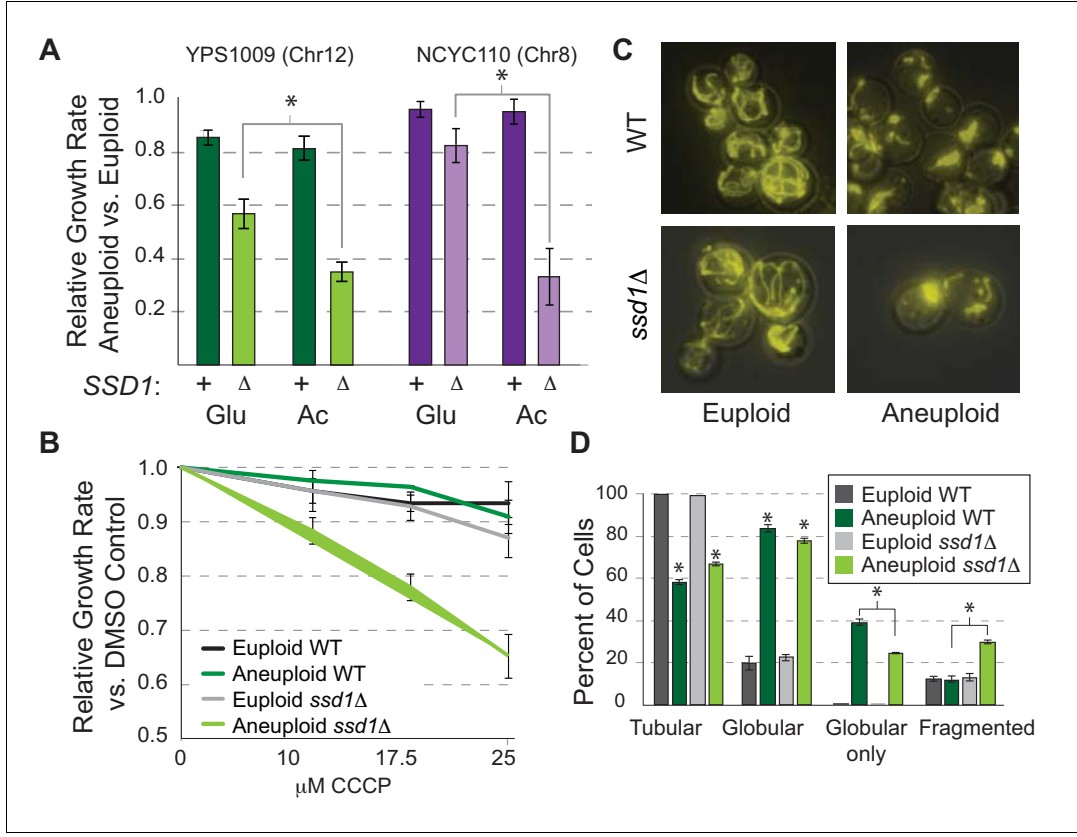

**Figure 4.** Ssd1 affects mitochondrial function and morphology. (**A**) Average and standard deviation of growth rates for denoted aneuploids versus euploids ± *SSD1* in glucose (Glu) or acetate (Ac). Asterisk, p<2e-4, replicate-paired T-test. (**B**) Average growth rates across CCCP doses. (**C**) Representative images of rhodamine-B stained mitochondria and D) quantified morphologies for cells with any tubular, any globular, only globular, or fragmented mitochondria (average and SEM, see Materials and methods). p<0.0001, Fisher's exact test.

The online version of this article includes the following figure supplement(s) for figure 4:

**Figure supplement 1.** No synergistic defects between aneuploidy and cell-wall or ER stress.

---

mitochondria fragmentation (*Figure 4D*). The West African aneuploid did not display globular mito-chondria but did display dysfunction (*Figure 4A*). Thus, *ssd1Δ* cells show numerous signs of mito-chondrial dysfunction compared to wild-type aneuploid cells.

We wondered if Ssd1 plays a direct role in mitochondrial function, perhaps by localizing nuclear-encoded mitochondrial mRNAs as it does cell-wall mRNAs. We first scored Ssd1 localization by cen-trifugation-based cellular fractionation. Ssd1 was reproducibly recovered in the organelle-enriched fraction that was depleted of cytosolic actin but enriched for markers of mitochondria (Cox2), ER (Dpm1), and vacuole (Vph1, *Figure 5A*), which themselves interact. Attempts to separate the com-ponents by immunoprecipitation of Ssd1 from the fractions were not successful. We next followed cellular localization of *MMR1* transcript, encoding a bud-mitochondria localized protein involved in mitochondrial inheritance (*Itoh et al., 2004*), by single-molecule RNA FISH (smFISH, *Figure 5B*). In most wild-type cells, mother-encoded *MMR1* was directed to the nascent bud, before the nucleus migrated. Although scoring precise differences in mRNA patterns was challenging, YPS1009_Chr12 *ssd1Δ* cells displayed twice as many buds lacking *MMR1* as wild-type aneuploids (*Figure 5C*, p<0.02, Fisher's exact test). In addition, the mutant showed double the cells lacking mitochondria as indicated by Rhodamine staining (*Figure 5D*), consistent with a defect in mitochondrial inheritance. It is possible that the mutant suffers from delayed dynamics, rather than fully aberrant localization in individual cells. Other nuclear-encoded mitochondrial mRNAs bound by Ssd1 were too abundant to follow by smFISH and will require further delineation. Nonetheless, together our work shows that Ssd1 associates with organelle fractions that include mitochondria (*Figure 5A*), binds several

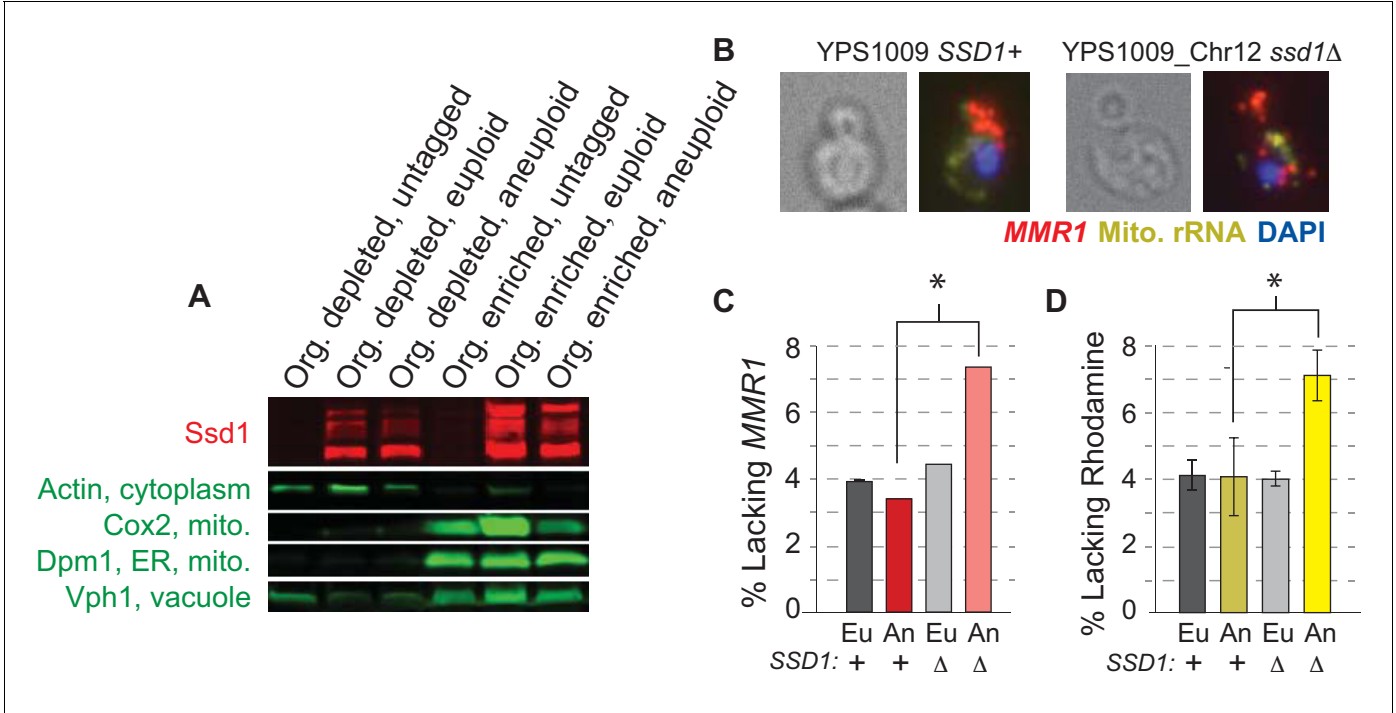

**Figure 5.** Ssd1 affects mRNA localization. (A) Representative Western blot showing Ssd1-GFP and markers of mitochondria, ER, and vacuole as detected in organelle-depleted and organelle-enriched fractions from *SSD1-GFP* or untagged-Ssd1 strains (see Methods). Ssd1 fragments migrating below the expected top band emerge during the fractionation incubations. (B) Representative smFISH showing bud-localized M*MR1* transcript in wild-type and *ssd1Δ* aneuploids. (C) Quantification of percent buds lacking *MMR1* from smFISH (see Materials and methods). (D) Percent of cells lacking Rhodamine staining. Histograms represent average and SEM across biological triplicates; *, p<0.02, Fisher's exact test.

nuclear-encoded mitochondrial transcripts (*Supplementary file 2*), and can influence abundance of mitochondrial proteins (*Figure 3B*) or localization patterns of bound transcript (*Figure 5B–C*), consistent with the requirement of *SSD1* for proper mitochondrial function in aneuploid cells (*Figure 4*).

## Combinatorial mitochondrial dysfunction and proteostasis stress underlie aneuploidy sensitivity in *ssd1Δ* cells

A remaining question is why Ssd1 dependence and mitochondrial dysfunction are more severe in aneuploids. We reasoned that underlying *ssd1Δ* defects are exacerbated by effects of chromosome amplification. One candidate is proteome stress that may emerge from over-production of proteins from the amplified chromosome in the absence of Ssd1-dependent translational silencing, which could tax the proteostatic buffering capacity specifically in the mutant (*Donnelly and Storchová, 2015*; *Oromendia and Amon, 2014*; *Oromendia et al., 2012*). Many recent studies have revealed a connection between mitochondrial function and cytosolic proteome stress: defects in mitochondrial protein import induce cytosolic proteostatic defense mechanisms, and misfolded cytosolic proteins interact with and can even be cleared by mitochondria (*Qureshi et al., 2017*; *Ruan et al., 2017*; *Wrobel et al., 2015*). Furthermore, mitochondrial defects and cytosolic proteostasis stress co-emerge in neurological syndromes, aging, and aneuploidy (*Oromendia and Amon, 2014*; *Helguera et al., 2013*; *Ruan et al., 2017*; *Franco-Iborra et al., 2018*; *Kauppila et al., 2017*).

To test the model that synergistic dysfunction underlies aneuploidy sensitivity in *ssd1Δ* aneuploid yeast, we applied nourseothricin (NTC), among the aminoglycoside drugs that induce mistranslation and protein misfolding (*Ling et al., 2012*). We confirmed that NTC treatment increased the number of Hsp104-GFP foci in the aneuploid wild type, and discovered that CCCP produced an even stronger effect even though wild-type aneuploids grew well in the drug (*Figures 6* and *4B*). Wild-type aneuploids were slightly sensitive to NTC, but the mutant was significantly more sensitive, beyond the expected additivity of aneuploidy and NTC response, revealing a synergistic defect induced by

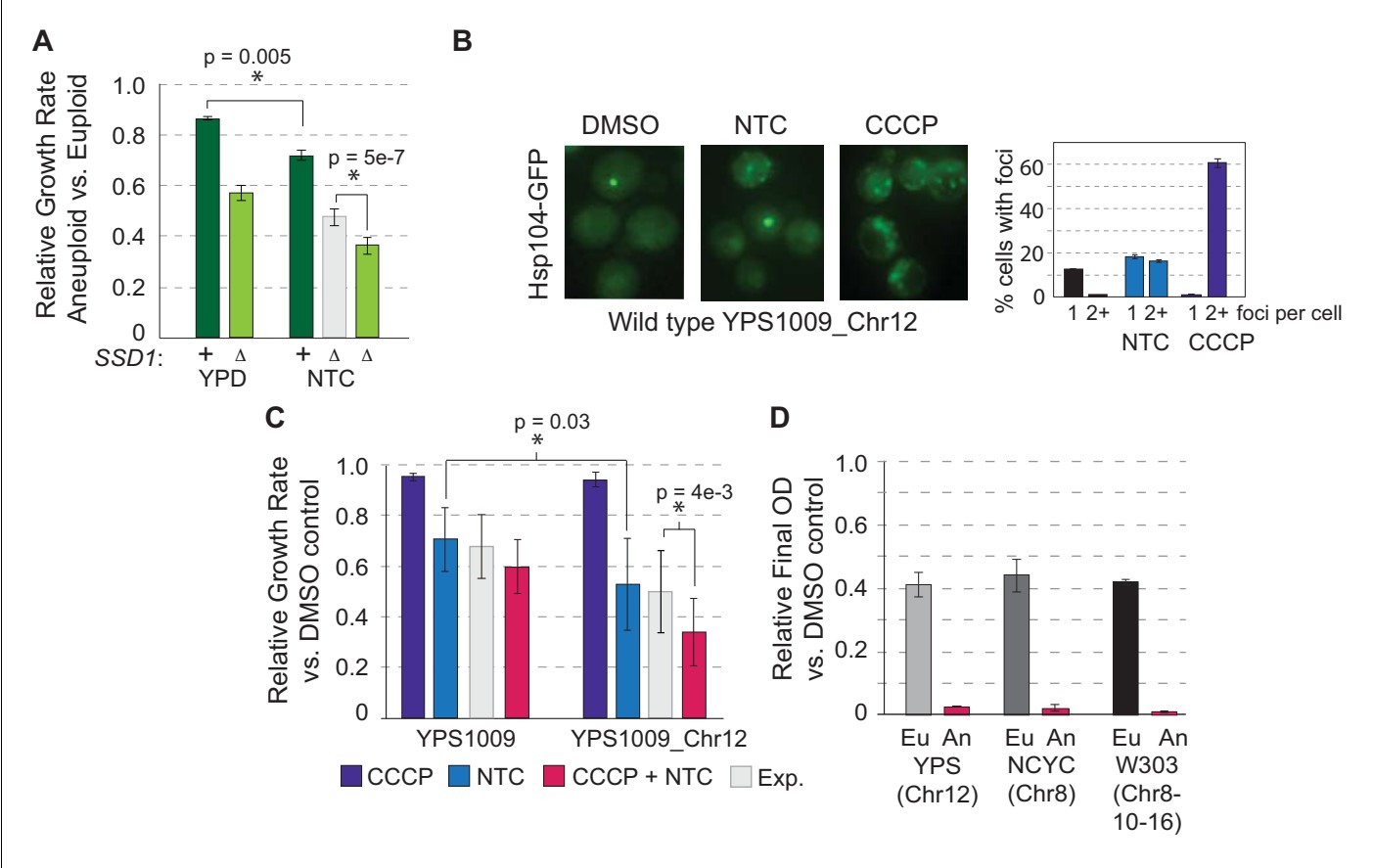

**Figure 6.** Protein misfolding and mitochondrial dysfunction sensitize aneuploids. (**A**) Average and standard deviation of relative growth rates in rich YPD medium or with 1 ug/mL NTC. The expected (Exp, light gray) additive effect was calculated based on the fold-drop in growth rate of NTC-treated euploid cells applied to the wild-type aneuploid growth rate in the absence of NTC. (**B**) Representative Hsp104-GFP foci triggered by 1 ug/mL NTC or 25 uM CCCP and quantification in YPS1009_Chr12 (average and SEM). (**C**) Average relative growth rate over three generations for indicated treatments or additive expectation (Exp, paired T-test). (**D**) Relative final optical density after overnight CCCP + NTC treatment in euploid (Eu) and aneuploid (An) strains (see Materials and methods).

The online version of this article includes the following source data for figure 6:

**Source data 1.** Clustal Omega alignment of YPS1009 and seven other strains with truncated alleles.

the drug in combination with aneuploidy (*Figure 6A*). The NTC sensitivity suggests that wild-type aneuploids with full length *SSD1* can largely buffer proteostasis upon chromosome amplification but may exist near capacity. Mitochondrial defect, protein over-abundance, and mislocalized transcripts/ proteins resulting from *SSD1* deletion may simply push cells over the edge (see Discussion).

This raised an important prediction: if synergistic defects in mitochondrial function and proteostasis sensitize *ssd1Δ* cells to chromosome amplification, then combinatorial drug treatment to mimic these defects should selectively target wild-type aneuploids. In fact, this was the case: wild-type euploid and aneuploid cells tolerated short-term CCCP and NTC individually, but when combined aneuploid growth was significantly delayed beyond the euploid strain and the expectation of additive effects (*Figure 6C*). Longer-term combinatorial drug treatment limited growth of euploid YPS1009 but selectively blocked proliferation in YPS1009_Chr12 (*Figure 6D*). The effect was persistent across strains and chromosome amplifications: combinatorial treatment halted over-night growth of W303 with duplications of Chr8, Chr10, and Chr16 and NCYC110 carrying extra Chr 8 (although in this strain CCCP was actually protective against NTC toxicity in the euploid cells at the doses used)(*Figure 6D*).

## Discussion

Our work has several major implications for understanding the consequences of aneuploidy and how to modulate them. First, we resolve the discrepancy in the literature between wild and laboratory-strain responses to aneuploidy, by showing that mutation of a single gene explains the phenotypic difference among strains studied here. Many of the yeast phenotypes previously reported as signatures of aneuploidy, including proteostasis defects, metabolic defects, cell-cycle defects, and transcriptome response, can be caused by *SSD1* deletion or mutation as seen in the commonly used W303 lab strain. This result explains why wild yeast from our studies, other studied *S. cerevisiae* strains, and pathogenic fungi do not show major defects upon chromosome amplification – these fungi have mechanisms to tolerate extra chromosomes. Our results underscore the importance of studying multiple strain backgrounds to understand model organism biology. At the same time, although W303 is clearly a sensitized strain we highlight that many important insights have come from its dissection. We propose that integrating our results with past yeast and mammalian studies reconciles to a holistic view of eukaryotic aneuploidy physiology.

Our model posits that Ssd1's function in translational silencing and mitochondrial physiology enable aneuploidy tolerance in wild yeast. The wild aneuploid strains studied here do not show signs of metabolic or proteostatic stress under standard growth conditions. But mimicking *ssd1Δ* defects through combinatorial drug treatment sensitizes cells to extra chromosomes, showing that it is indeed combined dysfunction in mitochondria and proteome management that is responsible for aneuploidy sensitivity. We propose that, under normal conditions, wild aneuploids handle the extra chromosome by buffering the effects of gene/protein amplification – yet cells may exist close to their proteostatic buffering capacity. Additional stress on the proteostasis system, due to drugs or *SSD1* deletion, pushes cells beyond capacity, thereby limiting fitness.

How does Ssd1 fulfill this function? Ssd1 has a clear role in translational regulation: it localizes to P-bodies during times of stress, suppresses encoded protein abundance via direct RNA binding, and is linked to the reduction of polysomes in aged cells (*Jansen et al., 2009*; *Kurischko et al., 2011a*; *Hu et al., 2018*; *Wanless et al., 2014*; *Kurischko et al., 2011b*). Ssd1's role in mitigating proteostatic stress likely emerges via RNA binding, since the *ssd1*$^{W303}$ allele lacks the carboxyl-terminal RNA binding domain (*Uesono et al., 1997*). A remaining question is if mRNAs bound by Ssd1 are especially relevant to proteome homeostasis. *Brennan et al. (2019)* recently identified aggregation-prone proteins in aneuploid W303, including proteins encoded on and off the amplified chromosomes. The hypothesis was raised that aggregation may be a beneficial mode of protein-dosage compensation (*Brennan et al., 2019*). However, that aggregation is a hallmark of *ssd1* deficiency, which itself causes aneuploidy sensitivity, argues against a beneficial function of aggregates. Instead, it points to a protective role for Ssd1 in handling aggregation-prone proteins. Consistent with this notion, the set of 22 proteins most prone to aggregation across W303 aneuploids is enriched for proteins encoded by Ssd1-bound transcripts identified in our study (p=0.005, hypergeometric test). Furthermore, as a group, proteins encoded by Ssd1 targets are predicted to display substantially higher fractions of disordered regions (based both on median IUPred score compared to all proteins and the fraction of residues with scores > 0.5, Mann Whitney p<2e-16); the trends remain significant even after disordered cell-wall proteins are removed from consideration. Although details of Ssd1's function remain to be worked out, these results are consistent with a role for Ssd1 in regulating where and when mRNAs are translated to minimize aggregation and misfolding, and to enable normal cells to handle extra chromosomes.

Our results also reveal that Ssd1 affects mitochondrial physiology in aneuploid cells. Defects in mitochondrial function and cytosolic proteome management have long been linked, in neurological syndromes, aging, and even aneuploidy. Disruption of mitochondrial protein folding, import, and localization induces cytosolic protein stress and triggers cytosolic proteostasis systems (*Wrobel et al., 2015*; *Wang and Chen, 2015*; *Nargund et al., 2012*), consistent with our observation that CCCP induces cytosolic Hsp104 foci (*Figure 6B*). Conversely, clearance of misfolded cytosolic proteins relies on mitochondria: in addition to providing sufficient ATP for chaperone function, mitochondria can retain and import misfolded cytosolic proteins for sequestration and degradation (*Ruan et al., 2017*; *Zhou et al., 2014*). Extreme cytosolic misfolding, for example aggregated Huntington protein, perhaps consequently causes mitochondrial dysfunction (*Franco-Iborra et al., 2018*; *Ocampo et al., 2010*). Defects in these processes also co-occur in aneuploid syndromes, which

produce altered mitochondrial morphology and function and premature aging phenotypes (*Oromendia and Amon, 2014*; *Bambrick and Fiskum, 2008*; *Chang and Min, 2005*). It is possible that mitochondrial defects in *ssd1Δ* cells arise as a secondary consequence of Ssd1 dysfunction; however, that Ssd1 binds several nuclear-encoded mitochondrial mRNAs, controls protein abundance of several of them, and purifies with mitochondria-enriched fractions raises the possibility of a more direct function. Given its role in localizing cell-wall mRNAs to the bud neck during division, Ssd1 may play a broader role in localizing and/or handling mitochondrial mRNAs – we provide evidence for one, *MMR1*, which showed a defect in localization patterns consistent with a defect in mitochondrial inheritance.

Our model that normal wild strains can handle the stress of extra chromosomes but exist near their buffering capacity is compatible with results from other systems. Some, but notably not all, aneuploid mouse and human cell lines show indirect signs of proteome stress, including increased autophagy and sensitivity to 17-AAG that inhibits Hsp90 chaperone (which is also required for proper chromosome segregation, confounding interpretation *Chen et al., 2012*). However, not all aneuploid lines display these signatures (*Santaguida et al., 2015*; *Stingele et al., 2013*; *Stingele et al., 2012*; *Donnelly et al., 2014*; *Tang et al., 2011*) and observed phenotypes are reportedly weaker than seen in W303 aneuploids (*Brennan et al., 2019*), as predicted by our study. While some phenotypic differences may result from differences in the load and identity of the aneuploid chromosome, another possibility is that proteostasis stress is not a universal feature of aneuploid cells. Rather, it may reflect a variable response influenced by environmental, developmental, or genetic differences in mitochondrial/proteostatic buffering capacity across lines. It has long been known that trisomy 21 produces phenotypes of variable severity in Down syndrome (DS), implicating genetic modifiers that augment tolerance (*Antonarakis, 2017*). A recent proteomic study showed that proteomes of unrelated DS skin fibroblasts showed some commonalities, including down-regulation of nuclear-encoded mitochondrial proteins, while other responses (such as altered lysosome activity) were variable across unrelated individuals and may thus contribute to variable DS severity (*Liu et al., 2017*). It is possible that natural genetic variation in wild yeast strains also contributes to natural variation in aneuploidy sensitivity. Interestingly, a recent large-scale genome sequencing study reported at least five truncated *SSD1* alleles segregating in yeast populations (*Figure 6— source data 1*) (*Peter et al., 2018*). The power of yeast genetics provides an opportunity to identify other modifiers of aneuploidy tolerance.

Ssd1 is orthologous to human Dis3L2 (*Heinicke et al., 2007*), an RNA binding protein best characterized for its ability to degrade poly-uridylated RNAs targeted for decay by terminal-uridyl transferases (TUTases) (*Astuti et al., 2012*; *Chang et al., 2013*; *Lubas et al., 2013*; *Malecki et al., 2013*; *Morris et al., 2013*; *Ustianenko et al., 2013*). Ssd1 is thought to have lost its catalytic activity (perhaps concomitant with loss of TUTase enzymes from *S. cerevisiae*; *Uesono et al., 1997*; *Viegas et al., 2015*). Dis3L2 was first identified in the causal mapping of Perlman syndrome, characterized by cellular over-growth, and is also implicated in Wilms tumor (*Astuti et al., 2012*). Dis3L2 shares several features with Ssd1: both can localize to the cytosol and nucleus, both bind RNAs and interact with P-bodies, and ablation of both proteins produces protein inclusion bodies (*Astuti et al., 2012*; *Malecki et al., 2013*; *Mori et al., 2018*; *Liu et al., 2018*; *Thomas et al., 2015*). Dis3L2 is also implicated in apoptosis trigged by mitochondrial signals (*Liu et al., 2018*; *Thomas et al., 2015*). Remarkably, mutation of Dis3L2 is also linked to aneuploidy: knockdown of Dis3L2 actually increases chromosome instability, leading to chromosome loss and aneuploidy (*Astuti et al., 2012*). Dissecting its role in aneuploidy syndromes is an exciting avenue for future work.

# Materials and methods

**Key resources table**

| Reagent type (species) or resource | Designation | Source or reference | Identifier | Additional information |
|---|---|---|---|---|

*Continued on next page*

*Continued*

| Reagent type (species) or resource | Designation | Source or reference | Identifier | Additional information |
|---|---|---|---|---|
| Gene (kan$^r$) | kan$^r$ | Yeast Knockout Collection; Horizon Discovery | | kanMX |
| Gene (*Klebsiella pneumoniae*) | hph | pAG26; Goldstein AL, McCusker JH | | hphMX |
| Gene (*Streptomyces noursei*) | nat1 | pPKI | | natMX |
| Gtrain (*Saccharomyces cerevisiae*) | YPS1009 Mat a Euploid, hoΔ::HYG | this study | AGY731 | Haploid, available on request from the Gasch Lab |
| Strain (*Saccharomyces cerevisiae*) | YPS1009 Mat alpha Euploid, hoΔ::HYG | this study | AGY732 | Haploid, available on request from the Gasch Lab |
| Strain (*Saccharomyces cerevisiae*) | YPS1009_Chr12 Mat a Disome12, hoΔ::HYG | this study | AGY735 | Haploid, available on request from the Gasch Lab |
| Strain (*Saccharomyces cerevisiae*) | YPS1009_Chr12 Mat alpha Disome12, hoΔ::HYG | this study | AGY736 | Haploid, available on request from the Gasch Lab |
| Strain (*Saccharomyces cerevisiae*) | YPS1009 Mat a Euploid, hoΔ::HYG, ssd1Δ::KAN | this study | AGY1444 | Haploid, available on request from the Gasch Lab |
| Strain (*Saccharomyces cerevisiae*) | YPS1009_Chr12 Mat a Disome12, hoΔ::HYG, ssd1Δ::KAN | this study | AGY1445 | Haploid, available on request from the Gasch Lab |
| Strain (*Saccharomyces cerevisiae*) | YPS1009 Mat a Euploid, hoΔ::HYG, ssd1-Δ2 (KANMX removed) | this study | AGY1503 | Haploid, marker rescued for plasmid expression, available on request from the Gasch Lab |
| Strain (*Saccharomyces cerevisiae*) | YPS1009_Chr12 Mat a Disome12, hoΔ::HYG, ssd1-Δ2 (KANMX removed) | this study | AGY1517 | Haploid, marker rescued for plasmid expression, available on request from the Gasch Lab |
| Strain (*Saccharomyces cerevisiae*) | YPS1009 Mat a Euploid, hoΔ::HYG, SSD1-GFP-SSD1YPS1009-terminator-NATMX | this study | AGY1446 | Haploid, GFP tagged Ssd1, available on request from the Gasch Lab |
| Strain (*Saccharomyces cerevisiae*) | YPS1009_Chr12 Mat a Disome12, hoΔ::HYG, SSD1-GFP-SSD1YPS1009-terminator-NATMX | this study | AGY1447 | Haploid, GFP tagged Ssd1, available on request from the Gasch Lab |
| Strain (*Saccharomyces cerevisiae*) | YPS1009 Mat a Euploid, hoΔ::HYG, his3Δ::KAN | this study | AGY1504 | Haploid, his3 deletion enabling HIS3 selection, available on request from the Gasch Lab |
| Strain (*Saccharomyces cerevisiae*) | YPS1009_Chr12 Mat a Disome12, hoΔ::HYG, his3Δ::KAN | this study | AGY1505 | Haploid, his3 deletion enabling HIS3 selection, available on request from the Gasch Lab |
| Strain (*Saccharomyces cerevisiae*) | YPS1009 Mat a Euploid, hoΔ::HYG, his3Δ::KAN, ssd1Δ::KAN | this study | AGY1506 | Haploid, his3 deletion enabling HIS3 selection, available on request from the Gasch Lab |
| Strain (*Saccharomyces cerevisiae*) | YPS1009_Chr12 Mat a Disome12, hoΔ::HYG, his3Δ::KAN, ssd1Δ::KAN | this study | AGY1507 | Haploid, his3 deletion enabling HIS3 selection, available on request from the Gasch Lab |
| Strain (*Saccharomyces cerevisiae*) | YPS1009 Mat a Euploid, hoΔ::HYG, his3Δ::KAN, PET123-GFP-ADH1terminator-HIS3M × 6 | this study | AGY1513 | Haploid, available on request from the Gasch Lab |
| Strain (*Saccharomyces cerevisiae*) | YPS1009_Chr12 Mat a Disome12, hoΔ::HYG, his3Δ::KAN, PET123-GFP-ADH1terminator-HIS3M × 6 | this study | AGY1514 | Haploid, available on request from the Gasch Lab |

*Continued on next page*

*Continued*

| Reagent type (species) or resource | Designation | Source or reference | Identifier | Additional information |
|---|---|---|---|---|
| Strain (*Saccharomyces cerevisiae*) | YPS1009 Mat a Euploid, hoΔ::HYG, his3Δ::KAN, HSP104-GFP-ADH1terminator-HIS3M × 6 | this study | AGY1518 | Haploid, available on request from the Gasch Lab |
| Strain (*Saccharomyces cerevisiae*) | YPS1009_Chr12 Mat a Disome12, hoΔ::HYG, his3Δ::KAN, HSP104-GFP-ADH1terminator-HIS3M × 6/HSP104 | this study | AGY1519 | Haploid, available on request from the Gasch Lab |
| Strain (*Saccharomyces cerevisiae*) | YPS1009 Mat a Euploid, hoΔ::HYG, his3Δ::KAN, ssd1Δ::KAN, HSP104-GFP-ADH1terminator-HIS3M × 6 | this study | AGY1520 | Haploid, available on request from the Gasch Lab |
| Strain (*Saccharomyces cerevisiae*) | YPS1009_Chr12 Mat a Disome12, hoΔ::HYG, his3Δ::KAN, ssd1Δ::KAN, HSP104-GFP-ADH1terminator-HIS3M × 6/HSP104 | this study | AGY1521 | Haploid, available on request from the Gasch Lab |
| Strain (*Saccharomyces cerevisiae*) | d-YPS1009_Chr12.2n Euploid | Hose et al. | AGY613 | Diploid, available on request from the Gasch Lab |
| Strain (*Saccharomyces cerevisiae*) | d-YPS1009_Chr12.4n Aneuploid | Hose et al. | AGY614 | Diploid, available on request from the Gasch Lab |
| Strain (*Saccharomyces cerevisiae*) | d-YPS1009_Chr12.2n Euploid, ssd1Δ::KAN/ssd1Δ::KAN | this study | AGY1560 | Diploid, available on request from the Gasch Lab |
| Strain (*Saccharomyces cerevisiae*) | d-YPS1009_Chr12.4n Aneuploid, ssd1Δ::KAN/ssd1Δ::KAN | this study | AGY1561 | Diploid, available on request from the Gasch Lab |
| Strain (*Saccharomyces cerevisiae*) | W303 Mat a Euploid ade2-1 his3-11,15 leu2-3,112 trp1-1 ura3-1 can1-100 Gal+ ade16Δ::KAN | this study | AGY1387 | Haploid, available on request from the Gasch Lab |
| Strain (*Saccharomyces cerevisiae*) | W303_Chr12 Mat a Disome12 ade2-1 his3-11,15 leu2-3,112 trp1-1 ura3-1 can1-100 Gal+ ade16Δ::KAN/ade16Δ::HYG | this study | AGY768 | Haploid, available on request from the Gasch Lab |
| Strain (*Saccharomyces cerevisiae*) | W303 Mat a Euploid ADE2+ his3-11,15 leu2-3,112 trp1-1 ura3-1 can1-100 Gal+ ade16Δ::KAN | this study | AGY1388 | Haploid, available on request from the Gasch Lab |
| Strain (*Saccharomyces cerevisiae*) | W303_Chr12 Mat a Disome12 ADE2+ his3-11,15 leu2-3,112 trp1-1 ura3-1 can1-100 Gal+ ade16Δ::KAN/ade16Δ::HYG | this study | AGY1389 | Haploid, available on request from the Gasch Lab |
| Strain (*Saccharomyces cerevisiae*) | W303 Mat a Euploid ade2-1 his3-11,15 leu2-3,112 trp1-1 ura3-1 can1-100 Gal+ | this study | AGY103 | Haploid, available on request from the Gasch Lab |
| Strain (*Saccharomyces cerevisiae*) | W303_Chr8 Mat a Disome8 ade2-1 his3-11,15 leu2-3,112 trp1-1 ura3-1 can1-100 Gal+ | this study | AGY1495 | Haploid, available on request from the Gasch Lab |
| Strain (*Saccharomyces cerevisiae*) | W303_Chr8-15 Mat a Disome8,15 ade2-1 his3-11,15 leu2-3, 112 trp1-1 ura3-1 can1-100 Gal+ | this study | AGY1496 | Haploid, available on request from the Gasch Lab |

*Continued on next page*

*Continued*

| Reagent type (species) or resource | Designation | Source or reference | Identifier | Additional information |
| --- | --- | --- | --- | --- |
| Strain (*Saccharomyces cerevisiae*) | W303_Chr8-10-16 Mat a Disome8,10,16 ade2-1 his3-11,15 leu2-3,112 trp1-1 ura3-1 can1-100 Gal+ | this study | AGY1497 | Haploid, available on request from the Gasch Lab |
| Strain (*Saccharomyces cerevisiae*) | YPS1009xW303 (sp100) Mat alpha Disome12 trp1-1 ade16Δ::KAN HYG+ | this study | AGY1548 | Haploid, available on request from the Gasch Lab |
| Strain (*Saccharomyces cerevisiae*) | d-NCYC110 Euploid | Hose et al. | AGY729 | Diploid, available on request from the Gasch Lab |
| Strain (*Saccharomyces cerevisiae*) | d-NCYC110_ Chr8-4n Aneuploid | Hose et al. | AGY703 | Diploid, available on request from the Gasch Lab |
| Strain (*Saccharomyces cerevisiae*) | d-NCYC110 Euploid, ssd1Δ:: KAN/ssd1Δ::KAN | this study | AGY1493 | Diploid, available on request from the Gasch Lab |
| Strain (*Saccharomyces cerevisiae*) | d-NCYC110_Chr8-4n Aneuploid, ssd1Δ:: KAN/ssd1Δ::KAN | this study | AGY1494 | Diploid, available on request from the Gasch Lab |
| Strain (*Saccharomyces cerevisiae*) | KCY40 (or VC580) Euploid, hoΔ:: MFA$^{prom}$-HYGMX-NATMX | Hose et al. | AGY806 | Haploid |
| Strain (*Saccharomyces cerevisiae*) | KCY40 (or VC580) Disome8, hoΔ:: MFA$^{prom}$-HYGMX-NATMX | Hose et al. | AGY1105 | Haploid |
| Strain (*Saccharomyces cerevisiae*) | KCY40 (or VC580) Euploid, hoΔ::MFA$^{prom}$-HYGMX-NATMX, ssd1Δ::KAN | this study | AGY1385 | Haploid, available on request from the Gasch Lab |
| Strain (*Saccharomyces cerevisiae*) | KCY40 (or VC580) Disome8, hoΔ:: MFA$^{prom}$-HYGMX-NATMX, ssd1Δ::KAN | this study | AGY1386 | Haploid, available on request from the Gasch Lab |
| Strain (*Saccharomyces cerevisiae*) | YPS1009_Chr12 Mat a Disome12, hoΔ::HYG + pPKI | this study | AGY735 transformed with plasmid | Haploid, available on request from the Gasch Lab |
| Strain (*Saccharomyces cerevisiae*) | YPS1009_Chr12 Mat a Disome12, hoΔ::HYG, ssd1Δ::KAN + pPKI | this study | ABY1445 transformed with plasmid | Haploid, available on request from the Gasch Lab |
| Strain (*Saccharomyces cerevisiae*) | YPS1009_Chr12 Mat a Disome12, hoΔ::HYG, ssd1Δ::KAN + pJH1-SSD1-W303 | this study | ABY1445 transformed with plasmid | Haploid, available on request from the Gasch Lab |
| Strain (*Saccharomyces cerevisiae*) | YPS1009_Chr12 Mat a Disome12, hoΔ::HYG, ssd1Δ::KAN + pJH1-SSD1-YPS1009 | this study | ABY1445 transformed with plasmid | Haploid, available on request from the Gasch Lab |
| Strain (*Saccharomyces cerevisiae*) | d-NCYC110_Chr8-4n Aneuploid + pJH1 | this study | AGY703 tranformed with plasmid | Diploid, available on request from the Gasch Lab |
| Strain (*Saccharomyces cerevisiae*) | d-NCYC110_Chr8-4n Aneuploid, ssd1Δ:: KAN/ssd1Δ::KAN + pJH1 | this study | AGY1494 transformed with plasmid | Diploid, available on request from the Gasch Lab |
| Strain (*Saccharomyces cerevisiae*) | d-NCYC110_Chr8-4n Aneuploid, ssd1Δ:: KAN/ssd1Δ::KAN + pJH1-SSD1-W303 | this study | AGY1494 transformed with plasmid | Diploid, available on request from the Gasch Lab |
| Strain (*Saccharomyces cerevisiae*) | d-NCYC110_Chr8-4n Aneuploid, ssd1Δ:: KAN/ssd1Δ::KAN + pJH1-SSD1-YPS1009 | this study | AGY1494 transformed with plasmid | Diploid, available on request from the Gasch Lab |

*Continued on next page*

*Continued*

| Reagent type (species) or resource | Designation | Source or reference | Identifier | Additional information |
|---|---|---|---|---|
| Strain (*Saccharomyces cerevisiae*) | W303 Mat a Euploid ade2-1 his3-11,15 leu2-3,112 trp1-1 ura3-1 can1-100 Gal+ ade1::HIS3, lys2::KAN | Torres et al. | AGY487 | Haploid |
| Strain (*Saccharomyces cerevisiae*) | W303_Chr12 Mat a Disome12 ade2-1 his3-11,15 leu2-3,112 trp1-1 ura3-1 can1-100 Gal+ ade16::HIS3 ade16::KAN | Torres et al. | AGY488 | Haploid |
| Strain (*Saccharomyces cerevisiae*) | W303 Mat a Euploid ade2-1 his3-11,15 leu2-3,112 trp1-1 ura3-1 can1-100 Gal+ ade1::HIS3, lys2::KAN + pJH1 | this study | AGY487 transformed with plasmid | Haploid, available on request from the Gasch Lab |
| Strain (*Saccharomyces cerevisiae*) | W303_Chr12 Mat a Disome12 ade2-1 his3-11,15 leu2-3,112 trp1-1 ura3-1 can1-100 Gal+ ade16::HIS3 ade16::KAN + pJH1 | this study | AGY488 transformed with plasmid | Haploid, available on request from the Gasch Lab |
| Strain (*Saccharomyces cerevisiae*) | W303_Chr12 Mat a Disome12 ade2-1 his3-11,15 leu2-3,112 trp1-1 ura3-1 can1-100 Gal+ ade16::HIS3 ade16::KAN + pJH1-SSD1-YPS1009 | this study | AGY488 transformed with plasmid | Haploid, available on request from the Gasch Lab |
| Antibody | Rabbit polyclonal Anti-GFP | Abcam | Abcam catalog #ab290 | Rabbit polyclonal; 1:2000 |
| Antibody | Mouse monoclonal Anti-Actin | Thermo Fisher Scientific | Thermo Fisher Scientific catalog #MA1-744 | Mouse monoclonal; 1:1000 |
| Antibody | Mouse monoclonal Anti-COX2 | Abcam | Abcam catalog #ab110271 | Mouse monoclonal; 1:500 |
| Antibody | Mouse monoclonal Anti-DPM1 | Abcam | Abcam catalog #ab113686 | Mouse monoclonal; 1:250 |
| Antibody | Mouse monoclonal Anti-VPH1 | Abcam | Abcam catalog #ab113683 | Mouse monoclonal; 1:1000 |
| Recombinant DNA reagent | pXIPHOS | GenBank accession MG897154 | PAM sgRNA sequence (GAATCGAATG CAACCGGCGC) that targeted KanMX | Higgins et al., Wrobel et al. |
| Recombinant DNA reagent | pPKI | this study | AGB185 | CEN plasmid with the natMX selection marker. |
| Recombinant DNA reagent | pJH1 | this study | AGB090 | CEN plasmid derived from pKI that has natMX selection marker. pJH is equivalent to pKI except for a fragment of unexpressed DNA that was removed during generation. |
| Recombinant DNA reagent | pJH1-SSD1-YPS1009 | this study | | ORF + 1000 bp upstream and 337 bp downstream of SSD1 from YPS1009 genomic DNA. Plasmid has natMX selection marker |
| Recombinant DNA reagent | pJH1-SSD1-W303 | this study | | ORF + 1000 bp upstream and 337 bp downstream of SSD1 from aW303 genomic DNA. Plasmid has natMX selection marker |

*Continued on next page*

*Continued*

| Reagent type (species) or resource | Designation | Source or reference | Identifier | Additional information |
|---|---|---|---|---|
| Recombinant DNA reagent | Molecular Barcoded Yeast (MoBY) v2.0 ORF Library | other | obtained from Great Lakes Bioenergy Research Center (GLBRC) | Ho, CH. et al. A molecular barcoded yeast ORF library enables mode-of-action analysis of bioactive compounds. Nat. Biotech. 27 (**Holland and Cleveland, 2012**), 369–377 (2009). |
| Sequence-based reagent | *MMR1* FISH probes | Stellaris | | designed against MMR1 mRNA |
| Sequence-based reagent | Mitochondrial rRNA FISH probes | Stellaris | | designed against 15 s and 21 s rRNA |
| Peptide, recombinant protein | von Hippel-Lindau (VHL) tumor suppressor | Kaganovich et al. | Addgene catalog #21053 | Kaganovich D, Kopito R, Frydman J. Misfolded proteins partition between two distinct quality control compartments. Nature. 2008 Aug 28. 454 (7208):1088–95. |
| Peptide, recombinant protein | *Aequorea victoria* GFP (S65T) | Huh et al. | | Huh W, Falvo JV, Gerke LC, Carroll AS, Howson RW, Weissman JS, and O'Shea EK (2003) Global Analysis of Protein Localization in Budding Yeast Nature 425:686–691. |
| Commercial assay or kit | Mitochondrial Yeast Isolation Kit | Abcam | Abcam catalog #ab178779 | |
| Commercial assay or kit | Illumina TruSeq Total RNA Stranded | Illumina | Illumina catalog #20020597; previously RS-122–2203 | |
| Commercial assay or kit | NEBNext Ultra DNA Library Prep Kit for Illumina | New England Biolabs | NEB catalog #E7370L | |
| Commercial assay or kit | Yeast Mitochondrial Stain Sampler Kit | Thermo Fisher Scientific | Thermo Fisher Scientific catalog #Y7530 | |
| Chemical compound, drug | Nourseothricin-dihydrogen sulfate(clonNAT) | Werner BioAgents | Werner BioAgents catalog #5.005.000 | |
| Chemical compound, drug | 4',6-Diamidino-2-phenylindole, dihydrochloride (DAPI) | Thermo Fisher Scientific | Thermo Fisher Scientific catalog #PI62247 | |
| Chemical compound, drug | Carbonyl cyanide 3-chlorophenylhydrazone (CCCP) | Sigma-Aldrich | Sigma catalog #C2759 | |
| Chemical compound, drug | Radicicol, Humicola fuscoatra | A.G. Scientific | A.G. Scientific catalog #R-1130 | |
| Chemical compound, drug | GFP-Trap Magnetic Agarose | Chromotek | Chromotek catalog #gtma-20 | |

## Strains and plasmids

Strains used in this study are listed in the Resource Table. W303_Chr8, W303_Chr8-Chr15, and W303_Chr8-Chr10-Chr16 were generated using the method of **Chen et al. (2012)**, passaging 16 generations in 20 µg/mL radicicol (A.G. Scientific) and plating on 8 or 16 µg/mL fluconazole to select for Chr8 aneuploidy. Karyotype was determined by array-comparative genomic hybridization and sequencing. W303 strains shown in **Figure 1E** were grown in SC-his + G418 to maintain marked copies of Chr12 (or corresponding markers in the otherwise isogenic wild type **Torres et al., 2007**). In general, deletions were generated by homologous recombination of relevant makers (e.g. *KAN-MX* or *HIS3*) into the designated locus, followed by diagnostic PCR to confirm correct integration and absence of the target gene. Because *ssd1Δ* cultures lose extra chromosomes (perhaps simply due to overtaking of the culture by stochastic euploid revertants), deletions were generated in wild-type strains that were then crossed to YPS1009_Chr12 *ssd1Δ*, followed by tetrad dissection to isolate aneuploid spore clones with desired genotypes. In all cases, aneuploidy was confirmed and periodically checked through diagnostic qPCR of one or two genes on the affected chromosomes (*AAT1* and *SDH2*) normalized to a single-copy gene elsewhere in the genome (*ERV25 or ACT1*) –

normalized ratios close to two reflect gene duplication, and ratios between 1.2–1.8X indicated partial loss of aneuploidy in the cell population. GFP-tagged genes were generated by integrating a *GFP-ADH2terminator-HIS*3 cassette (*Huh et al., 2003*) via homologous recombination into strain series AGY1504-1507 in which *HIS3* was previously deleted by replacement with *KAN-MX* marker. In the case of *SSD1-GFP* strains, a cassette consisting of GFP followed by the native *SSD1* terminator, 337 bp downstream of *SSD1*$^{YPS1009}$, was generated by PCR sewing with the *NAT-MX* marker. In all cases, cloned or tagged genes were confirmed by sequencing. *SSD1*$^{YPS1009}$ plus 1000 bp upstream and 300 bp downstream was cloned into a pRS-derived CEN plasmid for complementation. Because YPS1009_Chr12 *ssd1Δ* cannot tolerate the 2-micron plasmid, the VHL-GFP gene plus promoter and terminator sequences were cloned from pESC-LEU-GFP-VHL (ADDGENE #21053) into a pRS-derived CEN plasmid. Human VHL cannot fold without cofactors but is typically cleared from cells through proteasome activity (*McClellan et al., 2005*). Accumulation of VHL-GFP foci is generally taken as an inability to clear misfolded proteins.

## Growth conditions

Unless otherwise noted, strains were cultured for ~3 generations into log phase in rich YPD medium at 30°C, with the exception of microscopy experiments where cells were grown in low-fluorescence synthetic-complete medium and imaged live. Induction of VHL-GFP was performed by growing cells in YP with 2% raffinose + 2% galactose for 4 hr. Wild-type strains shown in *Figure 6C–D* were grown over-night in log-phase before addition of 1 ug/mL nourseothricin (Werner BioAgents, Jena, Germany) or 25 uM CCCP (Millipore-Sigma, St. Louis, MO). Growth rates were calculated by exponentially fitting changes in optical density. Relative final OD in *Figure 6D* was measured in biological triplicate after 24 hr growth of YPS1009, NCYC110, and W303 strains exposed to 25 uM CCCP with 0.5 ug/mL (NCYC, W303) or 1 ug/mL (YPS1009) NTC, respectively. Aneuploidy was periodically verified through diagnostic qPCR as described above. Expected growth rates under an additive model were estimated based on the fold-defect in one condition (*e.g.* aneuploidy versus euploidy) multiplied by the fold-defect in a second condition (*e.g.* NTC sensitivity in the euploid); significant differences in observed versus expected data were assessed with replicate-paired T-tests. Unless otherwise noted, all studies used at least biological triplicates with data represented as the average and standard deviation (except count data in *Figures 2B–C*, *4D*, *5C–D* and *6B* in which average and standard error of the mean across biological replicates is shown).

## Bulk-segregant phenotyping and mapping

Haploid strain AGY736 (YPS1009_Chr12 Mat alpha *ho::HYGMX*) was crossed with AGY768 (W303_Chr12 Mat a) and the resulting diploid sporulated and dissected evenly on agar plates. Colony diameter after 72 hr was scored for 208 spores; 76 spores ranking in the smallest ~40% of the distribution were scored for their propensity to lose Chr12 within 20 culture generations of growth: each spore was passaged for 2 days in liquid YPD, after which genomic DNA was isolated and Chr12 abundance scored by diagnostic PCR as described above. Loss of Chr12 signal was taken as aneuploidy sensitivity (20 spores, Pool A1) whereas cells that maintained Chr12 signal were taken as enriched for aneuploid tolerant cells (40 spores, Pool B1). Spore sp100 (Mat alpha *ADE2 HIS3 LEU2 trp- URA3*) that was prototrophic for influential markers was selected, its aneuploidy status verified by qPCR, and it was backcrossed to AGY735 (YPS1009_Chr12 Mat a *ho::HYGMX*). 37 segregants were scored only for their propensity to lose aneuploidy after 2 days of passage, generating an aneuploidy-sensitive pool (10 spores, Pool A2) and a pool enriched for aneuploidy-tolerant strains (25 spores, Pool B2). A control cross of euploid hYPS1009 X euploid W303 was generated and phenotyped for colony size as above. 46 and 50 spores were taken as 'small' (colony diameter <437 square pixels) or 'large' (colony >591 square pixels) for Pool D and Pool F, respectively.

Each clone was grown to saturation, an equal volume of each culture pooled appropriately, and genomic DNA isolated (Qiagen, Germantown, MD) from ach pool. Pooled genomic DNA was sequenced using NEBNext Ultra DNA Library Prep Kit for Illumina on an Illumina HiSeq 2000 to an average of 20M 100 bp reads per pool. To avoid potential mapping biases, an artificial reference genome was created where single nucleotide polymorphisms (SNPs) between the two parental genomes were substituted for a third allele not present in either genome. Reads from sequenced pools were aligned to the artificial reference using bwa-mem (*Li and Durbin, 2010*). A pileup at

known parental SNPs was created using samtools (*Li et al., 2009*), and allele counts at each SNP were calculated. SNP positions were filtered to retain SNPs with at least 15X coverage, both parental alleles scored, and allele frequency between 0.1–0.9 to eliminate false signals during bulk segregant analysis. Bulk-segregant analysis was performed using MULTIPOOL (v 0.10.1) (*Edwards and Gifford, 2012*) run across ~60,000 SNPs in contrast mode using the default recombination fraction (3300 cM) with -N set to the number of segregants in the aneuploidy-sensitive/small-colony pool in each cross (A1 = 20; A2 = 10; D = 46). Potential QTLs were identified at loci where allele frequency varied the greatest between the two pools. *SSD1* was validated as the causal locus through gene deletions and complementation as shown in *Figure 1*. Sequencing data for each pool are available in the Short Read Archive (SRA) under access number PRJNA548343, and MULTIPOOL output files are available in *Supplementary file 1*, as described in *Edwards and Gifford (2012)*.

## RNA sequencing, RNA immunoprecipitation, and plasmid barcode sequencing

RNA-seq was done as previously described (*Jovaisaite et al., 2014*) using total RNA isolated from log-phase cultures. Illumina reads were mapped to the S288c genome substituted with SNPs from YPS1009, NCYC110, or W303 as called in *Sardi et al. (2018)*, using bwa-meme. In general, data represent the average of biological triplicate, with the exception of h-YPS1009 strains shown in *Figure 2* done in quadruplicate and W303_Chr8-Chr15 and W303_Chr8-Chr10-Chr16 done in duplicate. Replicates for each strain suite were paired on the same day, enabling replicate-paired statistical analysis, done in edgeR (*Robinson et al., 2010*). Genes in *Figure 2* were selected by considering both YPS1009_Chr12 *ssd1Δ* versus YPS1009_Chr12 wild type and NCYC110_Chr8 *ssd1Δ* versus NCYC110_Chr8 wild type. Hierarchical clustering was performed using Cluster 3.0 (*Eisen et al., 1998*) and visualized in Java Treeview (*Saldanha, 2004*). Functional enrichment of GO terms was performed using the program SetRank (*Simillion et al., 2017*). Activation of the UPR was inferred from enrichment of Hac1 targets among induced genes ($p<1e-4$, hypergeometric test, compiled in *Chasman et al., 2014*), and signatures of mito-CPR was indicated as up-regulation of Pdr3 targets including *CIS3* as reported in *Weidberg and Amon (2018)*. Sequencing data are available from the GEO database under accession number GSE132425. Processed data are also available in *Supplementary file 2*.

RNA-immunoprecipitation (RIP) was performed similar to previously described (*Jansen et al., 2009*) with the following modifications: Cell lysate was treated with RQ1 RNase-free DNase (Promega, Madison, WI) for 15 min at room temperature, an aliquot was removed as the input material, and RNA was immunoprecipitated using GFP-Trap Magnetic Agarose (Chromotek, Planegg-Martinsried, Germany) against Ssd1-GFP from euploid and aneuploid lysate for 1 hr at 4°C (due to Ssd1-GFP degradation with longer incubation). An identical procedure was performed with untagged YPS1009 cells as a mock-RIP. Recovered RNA was subjected to Illumina sequencing as described above. RIP-seq was performed in duplicate for euploid and for aneuploid cells; bound transcripts were identified through combined edgeR (*Robinson et al., 2010*) analysis of the four RIP-seq samples, contrasting RIP to input for each sample and then to mock-IP normalized to its own input. Bound transcripts were taken as those with FDR < 0.05 (*Supplementary file 2*). Sequencing data are available from the GEO database under accession number GSE132425.

The suite of YPS1009_Chr12 strains (AGY731, AGY735, AGY1503, AGY1517) was transformed with Moby 2.0 high-copy expression library (*Magtanong et al., 2011*) and an aliquot removed as the starting pool. Cells were grown in biological triplicate for five generations in YPD medium, plasmid DNA collected from the starting and ending pools, and barcodes sequenced as previously described (*Magtanong et al., 2011*). Pools were normalized by total barcode reads per sample and fitness costs taken as the log2(fold change) in barcode abundance after versus before outgrowth. Data in *Figure 2D* represent the distribution of replicate-averaged data.

## Proteomics

Cell pellets were resuspended in 6 M guanidine HCl and boiled for 5–10 min; proteins were precipitated with methanol up to 90%, spun 5 min at 15 K g, and resuspended in lysis buffer (8 M urea, 100 mM Tris, pH = 8.0, 10 mM TCEP, 40 mM chloroacetamide). Samples were diluted to 1.5 M urea and digested overnight at room temperature with LysC (Wako Chemicals, USA) and for 3 hr with trypsin

(Promega, USA) at 1:50 enzyme to protein ratio. Samples were desalted using Strata X columns (Phenomenex Strata-X Polymeric RP, USA). For LC-MS/MS, samples were resuspended in 0.2% formic acid and separated via reversed phase (RP) chromatography. 2 μg of tryptic peptides were injected onto a capillary RP column prepared in-house and packed with 1.7 μm diameter Bridged Ethylene Hybrid C18 particles as described in *Shishkova et al. (2018)*. Columns were installed onto Dionex nanoHPLC (Thermo, Sunnyvale CA) and heated to 50°C using a home-built column heater. Mobile phase buffer A was composed of water and 0.2% formic acid, mobile phase B - 70% ACN and 0.2% formic acid. Samples were separated over a 120 min gradient at flow rate of 325 nl/min. Peptide cations were converted into gas-phase ions via electrospray ionization and analyzed using a Thermo Orbitrap Fusion Lumos (Thermo, San Jose CA) mass spectrometer, according to the previously published methods (*Hebert et al., 2018*). Raw data were searched using MaxQuant (v. 1.6.1.0) against *Saccharomyces cerevisiae* database (SGD, downloaded 10.15.2018). Searches were performed using precursor mass tolerance of 27 ppm and a product mass tolerance of 0.3 Da. Proteins were identified and quantified via MaxLFQ using default settings with enabled 'Match between runs,' requiring LFQ ratio of 1, and MS/MS spectra not required for LFQ comparisons. Raw data are available in the PRIDE database (Project accession # PXD013847). Prior to publication, reviewers can access the files using the following credentials: Username: reviewer95858@ebi.ac.uk, Password: 6w9IaMi3. Processed data and a list of proteins shown in *Figure 3* are available in *Supplementary file 2*, and normalized protein abundance data are available in *Supplementary file 3*.

## Mitochondrial fractionation, microscopy and single-molecule smFISH

Organelle-enriched and -depleted fractions were generated for euploid (AGY1446) and aneuploid YPS1009_Ch12 *SSD1-GFP* (AGY1447) and untagged *SSD1* cells as a control, using Mitochondrial Yeast Isolation Kit (Abcam, Cambridge, United Kingdom) according to manufacturer protocol with slight modifications to minimize protein degradation. Western blots were developed using anti-GFP ab290 (Abcam), anti-Actin MA1-744 (Thermo Fisher Scientific), anti-Cox2 ab110271 (Abcam), anti-Vph1 ab113683 (Abcam), and anti-Dpm1 ab113686 (Abcam) on a Li-COR Odyssey instrument (Model 9120). Cells for microscopy were plated on plain or poly-L-Lysine coated slides and either single images (HSP104-GFP in *Figure 6*) or z-stack images (all other microscopy) every 0.5 μm were acquired with an EVOS FL Auto two equipped with an RFP EVOS light cube. Z-stacks were collapsed into a single image with EVOS software for publication. Mitochondria in *Figure 4C* were visualized with Rhodamine B Hexyl Ester (ThermoFisher, R648MP) according to manufacturer's protocol; images represent an overlay of the bright-field image onto the fluorescence image to highlight cell boundaries. Cells were scored by marking total cells in bright-field images and the scoring presence or absence of Rhodamine B Hexyl Ester signal. A minimum of 380 cells were scored per strain across three biological replicates. Very similar results were obtained tracking Pet123-GFP signal.

smFISH was performed as previously described (*Gasch et al., 2017*) except performed on an EVOS FL Auto two and with transcripts detected manually in FIJI (*Schindelin et al., 2012*). FISH probe sets were designed against *MMR1* (conjugated to Quasar 670) and mitochondrial 15 s and 21 s rRNAs (conjugated to Quasar 570, Stellaris, Middlesex, United Kingdom). Mitochondrial morphology in *Figure 3D* was quantified using mitochondrial rRNA probes, which produced images very similar to Rhodamine staining but enabled visualization independent of mitochondrial membrane potential. Morphology was scored in each cell manually using the multi-point tool in FIJI, recording the number of cells with any tubular, any globular, or only globular morphologies. Cells with fragmented mitochondria were defined as those with at least three discontinuous fragments from the tubular structure or having at least three fragments in addition to the largest globular focus. >200–400 cells were scored for all microscopy experiments and across multiple biological replicates per strain. *MMR1* localization was scored by identifying buds (scored as cells lacking DAPI or containing bar nuclei by DAPI staining) and scoring those either containing or lacking *MMR1* transcripts.

## Acknowledgements

We thank Mike Place and Kevin Myers for experimental support, and Christina Scribano, Beth Weaver, and members of the Gasch Lab for constructive discussions.

## Additional information

### Funding

| Funder | Grant reference number | Author |
|---|---|---|
| National Cancer Institute | R01CA229532 | Audrey P Gasch |
| U.S. Department of Energy | DE-SC0018409 | Joshua J Coon<br>Audrey P Gasch |
| National Institutes of Health | P41 GM108538 | Joshua J Coon |
| National Institutes of Health | T32 GM007133 | H Auguste Dutcher |
| National Institutes of Health | T32 HG002760 | DeElegant Robinson |
| National Science Foundation | GRFP | Leah E Escalante |

The funders had no role in study design, data collection and interpretation, or the decision to submit the work for publication.

### Author contributions

James Hose, Conceptualization, Formal analysis, Investigation, Methodology; Leah E Escalante, DeElegant Robinson, Investigation, Methodology; Katie J Clowers, Evgenia Shishkova, Methodology; H Auguste Dutcher, Formal analysis, Investigation; Venera Bouriakov, Investigation; Joshua J Coon, Supervision; Audrey P Gasch, Conceptualization, Formal analysis, Supervision, Funding acquisition, Project administration

### Author ORCIDs

Audrey P Gasch https://orcid.org/0000-0002-8182-257X

### Decision letter and Author response

Decision letter https://doi.org/10.7554/eLife.52063.sa1
Author response https://doi.org/10.7554/eLife.52063.sa2

## Additional files

### Supplementary files

• Supplementary file 1. Zipped file with MULTIPOOL output files, as described in the MULTIPOOL manual, comparing pools A1 versus B1, A2 versus B2, and D versus F as described in Materials and methods. Plots represent the W303 allele frequency in the aneuploidy-sensitive or small-colony pools (red) versus the aneuploidy-tolerant or larger-colony pools (blue).

• Supplementary file 2. Compiled information and data. Genes bound by Ssd1 (column 3), genes shown in *Figure 2A* (column 4), proteins shown in *Figure 3* (column 5), log2(fold change) in mRNA abundance for denoted strains (columns 6–42), log2(fold change) in protein abundance for denoted strains (columns 46–56).

• Supplementary file 3. Normalized absolute protein abundance for each sample, see Materials and methods.

• Transparent reporting form

### Data availability

Sequencing data for genetic mapping are available in the Short Read Archive (SRA) under access number PRJNA548343, and MULTIPOOL output files are available in Supplementary file 1. RNA and RNA Immunoprecipitation (RIP) sequencing data are available from the GEO database under accession number GSE132425, and processed data are also available in Supplementary file 2. Raw proteomic data are available in the PRIDE database (Project accession # PXD013847); processed data are available in Supplementary file 2, and normalized protein abundance data are available in Dataset 3.

The following datasets were generated:

| Author(s) | Year | Dataset title | Dataset URL | Database and Identifier |
|---|---|---|---|---|
| Evgenia Shishkova, Joshua J Coon | 2020 | Aneuploid yeast proteomes in wild-type and ssd1 strains. | https://www.ebi.ac.uk/pride/archive/projects/PXD013847 | PRIDE, PXD013847 |
| Hose | 2020 | DNA mapping data | http://www.ncbi.nlm.nih.gov/bioproject/?term=PRJNA548343 | NCBI BioProject, PRJNA548343 |
| Hose J | 2020 | The Genetic Basis of Aneuploidy Tolerance in Wild Yeast | https://www.ncbi.nlm.nih.gov/geo/query/acc.cgi?acc=GSE132425 | NCBI Gene Expression Omnibus, GSE132425 |

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
