## [Decision Letter]

**Acceptance summary:**

The paper sheds light on the complicated story of aneuploidy in yeast and tolerance to aneuploidy in different strains. By extension, it should have relevance to aneuploidy tolerance in different human cells, especially tumor cells. Of interest also will be the quantitative genetics methodology used to identify a private mutation that leads to the differences in anueploidy tolerance in yeast strains. It will be important for future studies to delineate the mechanisms of aneuploidy tolerance.

**Decision letter after peer review:**

Thank you for submitting your article "The genetic basis of aneuploidy tolerance in wild yeast" for consideration by *eLife*. Your article has been reviewed by three peer reviewers, and the evaluation has been overseen by a Reviewing Editor and Jessica Tyler as the Senior Editor. The following individuals involved in review of your submission have agreed to reveal their identity: David Gresham (Reviewer #1); Juan Lucas Argueso (Reviewer #2); Kerry Bloom (Reviewer #3).

The reviewers have discussed the reviews with one another and the Reviewing Editor has drafted this decision to help you prepare a revised submission.

Summary:

The authors study aneuploidy in yeast, investigating the differences between wild yeast and a commonly used laboratory strain in aneuploidy tolerance. The authors use QTL mapping to pinpoint a variant version of *SSD1* in the laboratory strain as the causative agent of aneuploidy intolerance in the laboratory strain. The authors suggest that this *SSD1* variant impacts mitochondrial function and that this combined with proteostasis stress contributes to aneuploidy intolerance.

As this issue has been a source of much discussion in the yeast field, a resolution to the strain differences in terms of stress defects would be welcome. However, the current manuscript does not formally show that the *SSD1* variant is the causative agent. Two of the reviewers were enthusiastic about the potential impact of the manuscript while the third found the discussion of the *SSD1* variant not novel as it has long been known that the W303 strain contains this variant and that it contributes to some of the differing phenotypes of the W303 strain.

Essential revisions:

1) The authors need to formally show that the ssd1 allele of W303 is causative for aneuploidy intolerance. The right way to do this is to engineer the W303 allele into an otherwise pure YPS1009 strain background. Moreover, the exact nature of the W303 allele is very difficult to understand from the manuscript as the authors simply refer to other papers that have shown it is a hypomorph. Looking at the alignment in Figure 6—figure supplement 1, it is hard to understand where the truncating mutation appears. Therefore, basing all subsequent analyses on a complete gene deletion doesn't seem justified. The authors should provide more information on the nature of the allele and the rationale for using a null allele, rather than the W303 allele for subsequent assays.

The authors discuss aneuploidy sensitive and euploid crosses, based on propensity to lose Chr12. Presumably they are also losing other chromosomes? The authors should comment on the potential consequences of having a duplicated rDNA region in these cells; it seems this could have major phenotypic consequences.

As *SSD1* is involved in many pathways (TOR pathway, cell cycle regulation), it is possible that the mitochondrial effects are indirect. Claims that the effects are direct need to be unambiguously supported.

2) The statistical methods and the methodologies need to be better described.

Examples of points of confusion are listed here:

In Figure 2D how are the fitness data normalized? The median fitness differs between genotypes suggesting that these are absolute rather than relative values, but the Materials and methods states otherwise.

A key result is the identification of the sp100 segregant spore. It is stated "we realized during tetrad dissection, that adenine auxotrophy influenced aneuploidy tolerance (Figure 1—figure supplement 2)". All that is shown in Figure 1—figure supplement 2 is doubling time and growth. What are the data that the difference in growth rate is due to aneuploidy? The authors cite Figure 1—figure supplement 1 for showing growth rates of one vs two copies of Chr12 (Results section). They state that the number of copies of the chromosome was determined by qPCR. Methodology for the qPCR is not apparent.

The morphological differences in mitochondria were difficult to evaluate. The authors refer to globular and tubular morphologies described in Materials and methods. However, the Materials and methods just state that the multipoint tool in FIJI was used and do not define how these morphologies were quantitated. Disturbingly, they state in the supplemental figure legend (Supplementary figure 6) that mitochondrial morphologies were difficult to score, but were different. Which is the case, and why weren't the tools presumably used in Figure 4, used for Supplementary figure 6.

3) The studies here are limited to aneuploidy in haploid cells. However, aneuploidy is less deleterious in diploids and as wild yeasts are diploid, the effects of aneuploidy and the impact of the W303 ssd1 variant should be tested in diploids, comparing *SSD1/SSD1* 2N and 2N+1 versus ssd1D/ssd1D 2N and 2N+1. Whatever results are found, this should be added as an important point in the Discussion.

4) The general conclusions need to be further discussed. Examples are listed here:

The authors find that aneuploidy exacerbates aneuploidy sensitivity, but don't provide an explanation as to why this is the case.

I think this is a great example of the potential limitation of drawing conclusions on the basis of studies in a single strain background. The authors could discuss that in their conclusion.

---

## [Author Response]

Summary:The authors study aneuploidy in yeast, investigating the differences between wild yeast and a commonly used laboratory strain in aneuploidy tolerance. The authors use QTL mapping to pinpoint a variant version of SSD1 in the laboratory strain as the causative agent of aneuploidy intolerance in the laboratory strain. The authors suggest that this SSD1 variant impacts mitochondrial function and that this combined with proteostasis stress contributes to aneuploidy intolerance.As this issue has been a source of much discussion in the yeast field, a resolution to the strain differences in terms of stress defects would be welcome. However, the current manuscript does not formally show that the SSD1 variant is the causative agent. Two of the reviewers were enthusiastic about the potential impact of the manuscript while the third found the discussion of the SSD1 variant not novel as it has long been known that the W303 strain contains this variant and that it contributes to some of the differing phenotypes of the W303 strain.

We thank the reviewers for their positive assessment of our work and their recognition of the importance of settling this debate to better understand eukaryotic responses to aneuploidy.

The revised manuscript now shows definitively that the W303 allele of Ssd1 explains aneuploidy sensitivity in W303: expressing the *SSD1^YPS1009^*allele in multiple strain backgrounds recovers the growth defect of *ssd1-* aneuploid cells, whereas expressing the *ssd1^W303^*allele provides either no complementation or partial complementation depending on strain background. Most importantly, expressing *SSD1^YPS1009^*in aneuploid W303 almost completely complements its growth rate (with a residual contribution from the adenine auxotrophy). Thus, there is no ambiguity that it is the W303 allele of Ssd1 that explains the sensitivity of W303 to extra chromosomes.

However, we also wish to clarify why this work is important. The importance is not that we identified the allele explaining a phenotypic difference, but rather that we show that results that were previously taken to represent eukaryotic responses to aneuploidy – results that have been used to make inferences about human biology including aneuploidy syndromes and cancers – are specific to a mutant yeast strain: wild aneuploids do not show any phenotypes previously reported in W303 unless *SSD1* is defective. We go on to use the clean *SSD1* deletion to understand the role of an RNA binding protein in aneuploidy tolerance, studying multiple different strain backgrounds. Together with our model, this work will have a significant contribution to the aneuploidy field, much beyond defining a strain-specific difference.

Essential revisions:1) The authors need to formally show that the ssd1 allele of W303 is causative for aneuploidy intolerance. The right way to do this is to engineer the W303 allele into an otherwise pure YPS1009 strain background. Moreover, the exact nature of the W303 allele is very difficult to understand from the manuscript as the authors simply refer to other papers that have shown it is a hypomorph. Looking at the alignment in Figure 6—figure supplement 1, it is hard to understand where the truncating mutation appears. Therefore, basing all subsequent analyses on a complete gene deletion doesn't seem justified. The authors should provide more information on the nature of the allele and the rationale for using a null allele, rather than the W303 allele for subsequent assays.

As described above, we now provide definitive proof through allele swaps that the YPS1009 allele of Ssd1 maximally complements the growth defects in multiple strain backgrounds (including W303), whereas the truncated W303 allele expressed in the same strains does not. Allele swap experiments are provided in new Figure 1E. We also augmented the text to better describe the W303 allele: “The locus spanned *SSD1*, encoding an RNA-binding protein and known to harbor a premature stop codon in W303 that deletes 44% of the protein including conserved RNA binding domains (27-29).”

After describing the genetic basis of aneuploidy tolerance, we set out to better understand how Ssd1 enables cells to tolerate chromosome amplifications. For this we focused on the clean *SSD1* deletion to best interpret mutant phenotypes. We highlight that wild-type aneuploids with functional *SSD1* show none of the phenotypes of aneuploid W303 unless Ssd1 is defective. Thus, there is no ambiguity that Ssd1 is critical for handling extra chromosomes in multiple strains. We believe our work (and future work that emerges from it) will have a major contribution to understanding why extra chromosomes can be toxic and how normal cells deal with that stress.

The authors discuss aneuploidy sensitive and euploid crosses, based on propensity to lose Chr12. Presumably they are also losing other chromosomes? The authors should comment on the potential consequences of having a duplicated rDNA region in these cells; it seems this could have major phenotypic consequences.

The phenotype we observe is that *ssd1-* cultures rapidly lose cells that harbor the chromosome duplication. At this point, we do not know if the *SSD1* mutant loses chromosomes at a higher cellular rate or if the extreme growth-rate advantage of stochastic euploid revertants fully explains loss of aneuploidy from the culture. This is something we are currently investigating. We clarified the text in several places to state that the culture loses the aneuploid cells, perhaps simply because euploid revertants rapidly take over the culture.

The rDNA locus on Chr12 cannot explain the phenotypes we observed, because those phenotypes are also seen when other chromosomes are duplicated. We tested strains harboring extra copies of several different chromosomes (including Chr8, Chr10, Chr15, and/or Chr16) and show that growth reduction, transcriptome response, and sensitivity to mitochondrial stress and NTC are all caused by *SSD1* deletion or mutation. Furthermore, the phenotypes we study were previously reported in W303 harboring extra of any of the 16 yeast chromosomes (Torres et al., 2007 and other papers). Thus, these phenotypes cannot be explained by the rDNA locus on Chr 12. We added a statement to this effect in the Results section.

As SSD1 is involved in many pathways (TOR pathway, cell cycle regulation), it is possible that the mitochondrial effects are indirect. Claims that the effects are direct need to be unambiguously supported.

We agree that until the complete mechanism is known the mitochondrial effect could be indirect; however, that Ssd1 binds nuclear-encoded mitochondrial transcripts, affects protein abundance from several of those transcripts, and purifies with mitochondria-enriched fractions raises the possibility of a more direct role. We clarified the text to this effect: “It is possible that mitochondrial defects in *ssd1-* cells arise as a secondary consequence of Ssd1 dysfunction; however, that Ssd1 binds several nuclear-encoded mitochondrial mRNAs, controls protein abundance of several of them, and purifies with mitochondria-enriched fractions raises the possibility of a more direct function.

2) The statistical methods and the methodologies need to be better described.Examples of points of confusion are listed here:In Figure 2D how are the fitness data normalized? The median fitness differs between genotypes suggesting that these are absolute rather than relative values, but the materials and methods states otherwise.

We apologize for the confusion. We spent considerable time providing very detailed Materials and methods including statistical analysis. We have now clarified the Materials and methods to state that, “Pools were normalized by total barcode reads per sample and fitness costs taken as the log_2_(fold change) in barcode abundance after versus before outgrowth.”

A key result is the identification of the sp100 segregant spore. It is stated "we realized during tetrad dissection, that adenine auxotrophy influenced aneuploidy tolerance (Figure 1—figure supplement 2)". All that is shown in Figure 1—figure supplement 2 is doubling time and growth. What are the data that the difference in growth rate is due to aneuploidy?

The key result is in Figure 1—figure supplement 2B, which shows that reintroducing the *ADE2* gene into W303 significantly improves growth of the aneuploid cells but not the euploids. Thus, there is a synergistic benefit only when these cells are aneuploid.

The authors cite Figure 1—figure supplement 1 for showing growth rates of one vs two copies of Chr12 (Results section). They state that the number of copies of the chromosome was determined by qPCR. Methodology for the qPCR is not apparent.

qPCR was performed as described: “... aneuploidy was confirmed and periodically checked through diagnostic qPCR of one or two genes on the affected chromosomes (*AAT1* & *SDH2*) normalized to a single-copy gene elsewhere in the genome (*ERV25 or ACT1*) – normalized ratios close to 2 reflect gene duplication, and ratios between 1.2-1.8X indicated partial loss of aneuploidy in the cell population.”

The morphological differences in mitochondria were difficult to evaluate. The authors refer to globular and tubular morphologies described in Materials and methods. However the Materials and methods just state that the multipoint tool in FIJI was used and do not define how these morphologies were quantitated. Disturbingly, they state in the supplemental figure legend (Supplementary figure 6) that mitochondrial morphologies were difficult to score, but were different. Which is the case, and why weren't the tools presumably used in Figure 4, used for Supplementary figure 6.

We apologize for the confusion. The globular versus tubular morphology is very distinctive, and thus mitochondria in these forms were scored manually for Figure 4, as we now clarify. As stated in the Materials and methods, “Cells with fragmented mitochondria were defined as those with at least 3 discontinuous fragments from the tubular structure or having at least 3 fragments in addition to the largest globular focus.”

Because the other strains do not have globular types, it was much harder to score differences in mitochondrial character for other strains. To be fair, we removed the original Supplementary figure 6 and reference to other strains’ mitochondrial morphology from the manuscript.

3) The studies here are limited to aneuploidy in haploid cells. However, aneuploidy is less deleterious in diploids and as wild yeasts are diploid, the effects of aneuploidy and the impact of the W303 ssd1 variant should be tested in diploids, comparing SSD1/SSD1 2N and 2N+1 versus ssd1D/ssd1D 2N and 2N+1. Whatever results are found, this should be added as an important point in the Discussion.

Strains used in the original manuscript represented both haploid and diploid strains: YPS1009 and W303 examples in the original manuscript were haploid, but NCYC110 is a diploid strain tetrasomic for Chr 8. We now add additional data showing that diploid YPS1009 tetrasomic for Chr12 is even more sensitive to loss of *SSD1* than haploid YPS1009 disomic for Chr12 (compare new Figure 1—figure supplement 5C to Figures 1D and 4A). Thus, aneuploidy is not less delirious in diploids (the reviewer may be thinking of diploid strains that are trisomic for extra chromosomes and thus carry a lower load). Ssd1 is required in both haploid and diploid aneuploids of multiple genetic backgrounds.

4) The general conclusions need to be further discussed. Examples are listed here:The authors find that aneuploidy exacerbates aneuploidy sensitivity, but don't provide an explanation as to why this is the case.

We reworked the Discussion to make our model and supporting results clearer:

“Our model posits that Ssd1’s function in translational silencing and mitochondrial physiology enable aneuploidy tolerance in wild yeast. […] Additional stress on the proteostasis system, due to drugs or *SSD1* deletion, pushes cells beyond capacity, thereby limiting fitness.”

We go on to explain how Ssd1 could play a role through translational regulation and possibly mitochondrial physiology. Ssd1-bound mRNAs encoded proteins with a high fraction of intrinsic disorder, and this set includes proteins prone to aggregation in aneuploid W303. Because we see relatively few phenotypes in euploid cells lacking Ssd1 (albeit with strain-specific and base-ploidy nuances, see new Figure 1—figure supplement 5), we propose that the added stress of chromosome amplification, which produces additional protein burden especially in the absence of Ssd1, is especially toxic to cells lacking Ssd1. Ongoing work in our lab is dissecting the mechanistic details.

I think this is a great example of the potential limitation of drawing conclusions on the basis of studies in a single strain background. The authors could discuss that in their conclusion.

We thank the reviewer for this suggestion, and we added a statement to this effect in Discussion.